# New Algorithms for Fully-Dynamic $k$-center with Outliers

**Junyu Huang** [1]  **Zhize Li** [2]  **Zhen Zhang** [3][4]  **Xujia Li** [5]  **Jianxin Wang** [1][6]  **Qilong Feng** [1]

## Abstract

In this paper, we study the fully dynamic $k$-center with outliers problem, where points are inserted and deleted over time and the goal is to maintain an approximate clustering while discarding up to $z$ outliers. Existing algorithms typically rely on radius guessing to maintain cluster representations, leading to update and query times that depend explicitly on the aspect ratio $\Delta$. We propose a layered-sampling framework that avoids radius guessing by maintaining a hierarchy of sampled structures, which can separate most inliers from potential outliers and refine the remaining uncertain points. The resulting algorithm achieves $\tilde{O}(k^2/\epsilon^4)$ update and query time independent of $\Delta$, while guaranteeing a constant-factor approximation with $(1 + \epsilon)z$ outliers discarded. Under mild assumptions, our result is complemented by a lower bound in metric space query model.

## 1. Introduction

Clustering is a fundamental problem in the field of unsupervised learning with various practical applications. Among different mathematical formulations, $k$-center clustering is one of the most widely studied. The goal of $k$-center clustering is to select a center set with size $k$ and assign each data point to its nearest center, while the maximum assignment distance of the data points is minimized. Over the past decades, the standard metric $k$-center problem has been extensively studied (Megiddo & Supowit, 1984; Feder & Greene, 1988), where several approximation schemes with tight theoretical guarantees have been proposed (Gonzalez,

1985; Hochbaum & Shmoys, 1985). However, with recent advances in large-scale data processing and online platforms, data to be clustered often arrives continuously and changes dynamically in many real-world applications. For example, data in transportation networks continually updates as vehicles move and passengers enter or leave the system. In such settings, the static clustering models may fail to support the efficient structural updates of the evolving data, which motivates the studies of fully-dynamic clustering models.

In the fully-dynamic setting, the clustering structure must be efficiently updated as the point set changes under a stream of insertions and deletions operations. The focus is on building a dynamic algorithm that has low update and query times. Along this line of research, a series of fully-dynamic algorithms for $k$-center clustering have been proposed. Chan et al. (2018) formulated the fully-dynamic $k$-center problem and proposed the first approximation algorithm with sub-linear update time. To handle point updates while guaranteeing approximations on clustering quality, the proposed algorithm maintains a ball-coverage structure with a uniform covering radius. By enumerating candidate radii within a range defined by approximate upper and lower bounds of pairwise distances, it achieves a $(2 + \epsilon)$-approximation with $O(k^2 \log(\Delta)/\epsilon)$ update time and $O(k)$ query time, where $\Delta$ denotes the aspect ratio of the clustering data (defined as the ratio between the maximum and minimum pairwise distances). Following this work, several results have been proposed with improved memory (Chan et al., 2020), or improved update time in special metric spaces such as doubling metrics (Gan & Golin, 2024; Goranci et al., 2021; Pellizzoni et al., 2023). Very recently, a fully dynamic $(2 + \epsilon)$-approximation algorithm was proposed (Bateni et al., 2023), where an amortized update time of $O(k \cdot \text{polylog}(\Delta, n))$ can be achieved. This result nearly matches the lower bound update time of $\Omega(k)$ for the fully-dynamic $k$-center problem, as shown in the literature (Bateni et al., 2023).

Although existing fully-dynamic $k$-center algorithms can achieve near-optimal update time, a recurring issue for dynamic data clustering is the presence of outliers. Such outliers arise naturally in practice due to measurement noise, corrupted or incomplete records, and irregular user-generated inputs. In fully-dynamic setting, frequent insertions and deletions may also generate short-lived or unrepresentative points that naturally behave as outliers. Even

[1] School of Computer Science and Engineering, Central South University, China [2] School of Computing and Information Systems, Singapore Management University, Singapore [3] School of Advanced Interdisciplinary Studies, Hunan University of Technology and Business [4] Xiangjiang Laboratory, Changsha, China [5] Hong Kong University of Science and Technology, Hong Kong SAR, China [6] The Hunan Provincial Key Lab of Bioinformatics, Central South University, Changsha 410083, China. Correspondence to: Qilong Feng <csufeng@mail.csu.edu.cn>.

*Proceedings of the 43rd International Conference on Machine Learning*, Seoul, South Korea. PMLR 306, 2026. Copyright 2026 by the author(s).

the existence of a small fraction of outliers can significantly deteriorate the clustering results, forcing centers to shift toward extreme positions and thereby degrading the overall clustering quality. This motivates recent studies of the fully-dynamic clustering settings in the presence of outliers.

Inspired by the classical $k$-clustering with outliers model (Charikar et al., 2001; Bhaskara et al., 2019; Ding et al., 2019; Chakrabarty et al., 2020; Grunau & Rozhoň, 2022), Chan et al. (2022) formulated the fully-dynamic $k$-center with outliers problem, which is also referred to as the fully-dynamic $(k, z)$-center problem. In this setting, the input consists of an initial point set $P$, together with parameters $k \in \mathbb{N}$ and $z \in \mathbb{N}$, denoting the number of clusters and the outlier budget. The focus is on designing a dynamic algorithm with low update time (ideally independent of data size) for both insertions and deletions. Additionally, upon receiving a query, a center set $C \subseteq P$ is required to be returned such that by discarding at most $z$ data points in $P$, the maximum distance from the remaining data points to their nearest points in the center set $C$ is minimized.

For the fully-dynamic $(k, z)$-center problem, a lower bound update time of $\Omega(z)$ has been established for achieving any $O(1)$-approximate solution when exactly $z$ outliers are allowed to be discarded (Chan et al., 2024). However, in many practical settings, outliers often constitute a non-negligible fraction of the data. In this regime, the $\Omega(z)$ lower bound implies an $\Omega(n)$ worst-case update time, which may significantly influence the scalability of the dynamic algorithms. Therefore, to achieve sub-linear update time, it is natural to relax the outlier budget constraint and allow slightly more than $z$ outliers to be discarded. Following this setting, recent studies have focused on algorithms with bi-criteria approximations (i.e., with slightly more than $z$ outliers discarded), thereby avoiding the update time dependence on the outlier budget $z$. Chan et al. (2022) proposed the first bi-criteria result with $(14+\epsilon)$-approximation on clustering quality and $(1+\epsilon)z$ outliers discarded, where the update time and query time are $O(\frac{k^2}{\epsilon^3} \log \Delta)$ and $O(\frac{k^2 \log^2 k \log \log \Delta}{\epsilon^3})$, respectively. The proposed algorithm works by dynamically maintaining ball-coverage structures with uniform radius ranging from $d_{\min}$ to $d_{\max}$ (obtained from prior knowledge), where $d_{\min}$ and $d_{\max}$ are the minimum and maximum pairwise distance of the clustering data, respectively. This guarantees that the candidate range contains a radius that well approximates the optimum, and balls of this radius can be used to construct weighted instances for answering queries and producing the final clustering solutions with provable guarantees. To further improve the approximation guarantee and query efficiency, Biabani et al. (2024) proposed a leveling coverage strategy that directly maintains exactly $k$ clusters (or balls) instead of weighted instances. This avoids the two-stage approximation loss from instances compressions, leading to improved clustering guarantees and faster

query time. With leveling coverage strategy, an improved $(4 + \epsilon)$-approximation can be achieved with $(1 + \epsilon)z$ outliers discarded, where the update time and query time are $O(\epsilon^{-3}k^6 \log(k) \log(\Delta))$ and $O(k)$, respectively.

A separate line of research considers metric spaces with bounded doubling dimension (Biabani et al., 2023; De Berg et al., 2023; Pellizzoni et al., 2023), aiming to achieve better approximation and faster update time. In this setting, by integrating geometric data structures such as navigating nets, it allows to maintain a single-criterion constant approximation with running time independent of the outlier budget $z$. For a fixed doubling dimension ddim, the best known result (Biabani et al., 2023) achieves a $(3 + \epsilon)$-approximation while discarding exactly $z$ outliers, with update time $\frac{1}{\epsilon^{O(\text{ddim})}} \log n \log \Delta$ and query time $\frac{1}{\epsilon^{O(\text{ddim})}} k \log n \log \log \Delta$. However, in the worst cases, these approaches may have exponential dependence on the dimension for both the update and query times, which may deteriorate the performances in high-dimensional settings.

While existing algorithms for the fully-dynamic $(k, z)$-center problem achieve sub-linear update and query time in the data size, their theoretical guarantees typically rely on additional assumptions. Specifically, these include: (1) prior knowledge of the maximum and minimum pairwise distances to maintain the radius searching range; and (2) an explicit dependence on the aspect ratio $\Delta$ in both update and query times. These requirements naturally come from a basic characterization of the $k$-center objective: the optimal solution is determined by the smallest radius under which the points can be covered by $k$ balls. Having prior knowledge of the optimal clustering radius can simplify the algorithm design and improve efficiency through greedy ball coverage strategies. An efficient way for optimal radius estimation is to enumerate candidate radii from a range defined by the minimum and maximum pairwise distances. However, maintaining such bounds under fully-dynamic settings is itself nontrivial. While an approximate upper bound for the maximum pairwise distances can be estimated efficiently via random projection techniques (Cohen-Addad et al., 2021), maintaining a meaningful lower bound on the minimum pairwise distance is far more challenging. Even in static settings, estimating the minimum pairwise distance without inspecting a near-quadratic number of pairs is highly nontrivial. This makes radius-range maintenance a major bottleneck when no prior bounds are available.

Even with prior knowledge of distance bounds, a further encountered issue is the explicit dependence on the aspect ratio $\Delta$ in both update and query time, which mainly arises from the radius-guessing process. For the fully-dynamic $(k, z)$-center problem, sub-linear time bounds are typically achieved only when $\Delta$ is bounded by a polynomial function of the data size, or when the metric has bounded doubling

dimension. As pointed out in the literature (Bhattacharjee & Moshkovitz, 2021), these assumptions can be restrictive, where a bounded aspect ratio does not cover natural inputs generated from mixture models. In the worst cases, the aspect ratio $\Delta$ can be arbitrarily large (Bhattacharjee & Moshkovitz, 2021; Nguyen et al., 2022), which may limit the scalability of algorithms whose update or query time depends on $\Delta$. Moreover, in static settings, the $\Delta$-dependence can often be reduced by using a binary-search strategy over candidate radii, which improves the overhead from $O(\log \Delta)$ to $O(\log \log \Delta)$. However, the binary searching strategy does not generally extend to dynamic updates. There is no available deterministic procedure that can support binary search for improving the update time. Therefore, a central challenge in approximation algorithms design for the fully-dynamic $(k, z)$-center problem with improved time bounds is to minimize the impact of the aspect ratio $\Delta$.

On the other hand, for $k$-clustering problems (such as $k$-means and $k$-median) in the static setting with outliers (Chen et al., 2018; Deshpande et al., 2020; Huang et al., 2025) or the fully-dynamic setting (Bhattacharya et al., 2023) (without outliers), there are algorithms that can achieve $\Delta$-independent time bounds using sampling-based approaches. Specifically, these approaches usually adapt a continuous sampling strategy, which iteratively covers a certain fraction of points instead of using fixed clustering radius. In this way, the clustering radius can be automatically determined during the covering process, which avoids the radius guessing process. These results shed some light on how sampling can be used to remove the aspect-ratio dependence. However, a key challenge is to bound the number of points discarded as outliers. During the sampling and covering process, once the size of uncovered data points becomes comparable to the outlier budget, the uncovered set is no longer dominated by true inliers. As a result, the sampling procedure may discard significantly more than $z$ outliers, or require a sample size that grows with the data size to preserve approximation guarantees. Even in static $(k, z)$-center setting, existing sampling-based approaches typically achieve a constant approximation only when a large fraction of the uncovered points (i.e., $O(z)$ points) are discarded as outliers (Chen et al., 2018). Consequently, it remains unclear how to adapt these sampling-based ideas to ensure that no more than $(1 + \epsilon)z$ points are discarded as outliers in the static setting, and how to further generalize them to the fully dynamic setting with insertions and deletions.

## 1.1. Our Contributions

This paper focuses on the fully-dynamic $(k, z)$-center problem in general metric spaces. The main objective is to design algorithms that can achieve efficient update and query times, while maintaining constant approximation with only $(1+\epsilon)z$ outliers discarded. In this setting, existing approaches typ-

ically assume prior knowledge on the bounds for pairwise distances and rely on radius-guessing strategy to maintain approximate solutions. Although effective, this usually introduces an $O(\log \Delta)$ overhead in the update or query time (Biabani et al., 2024; Chan et al., 2022), which can become dominant when the aspect ratio $\Delta$ is large.

Our main contribution is a new layered-sampling approach that can achieve $\Delta$-independent update and query time, while removing any prior assumptions on pairwise-distance bounds. The intuitive idea behind is to maintain a sequence of layered structures via a greedy coverage strategy, where each layer selects representatives from the currently uncovered points and then peels off the points that are close to these representatives. Rather than fixing a covering radius in advance, each structured layer induces its own radius through the representatives it selects and the covered points they capture. Therefore, the algorithm can avoid the radius guessing process during the updates. To obtain a sharper bound on the discarded outliers, we introduce a stage-wise construction scheme that builds the layered structure progressively. At the very beginning, the initial layers consist of small samples, which aims to cover most inliers. After this stage, a transition layer is constructed to obtain high-quality representatives for the remaining uncovered points, by slightly increasing the representative sample size. Finally, a fine-grained division stage is used to construct subsequent layers that recover the discarded inliers in small batches, thereby ensuring that no more than $\epsilon z$ inliers are missed. These components together ensure that at most $(1 + \epsilon)z$ points are discarded as outliers when receiving a query.

To guarantee sub-linear update time, an update-delay rule is adapted to avoid repairing the layered structure after every single insertion or deletion. Instead, updates are recorded by counters, and the maintained layered structures are rebuilt only when the accumulated point changes reach a certain threshold. This spreads the reconstruction cost over a series of updates and thus yields sub-linear update time.

Upon receiving a query, the layered structures are used to construct a series of weighted point sets, referred to as summaries, that can effectively represent the original dataset. Queries are answered by executing an existing $(k, z)$-center routine on the resulting weighted summaries. To further eliminate the aspect ratio dependency on the query time, a greedy pre-processing is applied to merge representatives that are close to each other in the summaries. With this step, the remaining representatives exhibit a separation property. Under this structure, either an approximate optimal radius is reflected by a pairwise distance between representatives and can be identified by a binary-search procedure over these distances, or the top-$k$ representatives with the largest weights already capture almost all inliers and directly yield a feasible solution. In both cases, the query can be answered

*Table 1.* Comparisons with previous fully-dynamic $(k, z)$-center algorithms, where $k$ is the number of clusters, $z$ is the outlier budget, $\Delta$ is the aspect ratio of the dynamic data, ddim is the doubling dimension, $\epsilon$ is used to control number of additionally discarded outliers.

| Metric | Update Time | Query Time | Approximation | Outliers Discarded | Ref |
|---|---|---|---|---|---|
| Doubling | $\frac{1}{\epsilon^{O(\text{ddim})}} \log \Delta$ | $\frac{1}{\epsilon^{O(\text{ddim})}} (k+z)^2 \log \Delta$ | $2 + \epsilon$ | $z$ | Pellizzoni et al. (2023) |
| Doubling | $O((\frac{k}{\epsilon^{\text{ddim}}} + z) \log^4(\frac{k\Delta}{\epsilon\delta}))$ | $O((\frac{k}{\epsilon^{\text{ddim}}} + z)^2 k \log(\frac{k}{\epsilon^{\text{ddim}}} + z))$ | $3 + \epsilon$ | $z$ | De Berg et al. (2023) |
| Doubling | $\frac{1}{\epsilon^{O(\text{ddim})}} \log n \log \Delta$ | $\frac{1}{\epsilon^{O(\text{ddim})}} k \log n \log \log \Delta$ | $3 + \epsilon$ | $z$ | Biabani et al. (2023) |
| General | $O(\frac{k^2}{\epsilon^3} \log \Delta)$ | $O(\frac{k^2 \log^2 k \log \log \Delta}{\epsilon^3})$ | $14 + \epsilon$ | $(1 + \epsilon)z$ | Chan et al. (2022) |
| General | $O(\epsilon^{-3} k^6 \log k \log \Delta)$ | $O(k)$ | $4 + \epsilon$ | $(1 + \epsilon)z$ | Biabani et al. (2024) |
| General | $\tilde{O}(k^2/\epsilon^4)$ | $\tilde{O}(k^2/\epsilon^4)$ | $O(1)$ | $(1 + \epsilon)z$ | Ours |

without relying on $\Delta$-dependent radius searching. Putting all these together, we have the following results.

**Theorem 1.1.** *There exists a randomized algorithm for the fully-dynamic $(k, z)$-center problem such that with constant probability, an $O(1)$-approximate solution can be returned with $(1+\epsilon)z$ outliers discarded, where the amortized update time is $\tilde{O}(k^2/\epsilon^4)$[1] and the query time is $\tilde{O}(k^2/\epsilon^4)$.*

To further enhance the scalability, under mild assumptions on optimal cluster sizes, we also present a faster fully-dynamic $(k, z)$-center algorithm via data compression techniques. The key strategy behind is to compress the data size through independent sampling during the updates, where each point is sampled with probability independent of data size. This further reduces the data size by a factor of $z$, while introducing only $O(1)$ approximation loss. We further complement this result with a lower bound showing that, in the general metric space query model, any algorithm with $O(1)$-approximation and $O(z)$ outliers discarded must incur $\Omega(k^2/z)$ update time. This indicates that our results are close to the best possible bound under this model.

**Theorem 1.2.** *Under the assumption that each optimal cluster has size $\Omega(z)$, there exists a randomized algorithm for the fully-dynamic $(k, z)$-center problem such that with constant probability, an $O(1)$-approximate solution can be returned with $O(z)$ outliers discarded, where the amortized update time is $\tilde{O}(k^2/z)$ and the query time is $\tilde{O}(k^2/z)$.*

**Theorem 1.3.** *Any randomized algorithm for the fully-dynamic $(k, z)$-center problem in the setting $k \geq C$, $z \geq Ck \log k$, and $n \geq Cz$ for an absolute constant $C$ that with probability at least 0.5 gives an $O(1)$ approximation with $O(z)$ outliers discarded in the general metric space query model, requires $\Omega(k^2/z)$ update time.*

Table 1 presents the comparisons with previous fully-dynamic $(k, z)$-center algorithms. It can be seen from the table that, our proposed method is the first to achieve both update and query times that are independent of the aspect ratio $\Delta$, while maintaining comparable guarantees on clus-

tering quality with only $(1 + \epsilon)z$ outliers discarded.

To summarize, the main contributions are as follows.

- We propose a layered-sampling method for the fully-dynamic $(k, z)$-center problem that eliminates the aspect ratio dependency on both the update and query time. The proposed method makes no assumption on pairwise distance bounds and can guarantee a constant-factor approximation with $(1 + \epsilon)z$ outliers discarded.

- Under mild assumptions on optimal cluster sizes, we propose a faster fully-dynamic $(k, z)$-center algorithm with both $\tilde{O}(k^2/z)$ update and query time, guaranteeing a constant-factor approximation with $O(z)$ discarded outliers. This is complemented by a lower bound of $\Omega(k^2/z)$ on the update time in the general metric query model.

## 2. Preliminaries

In this section, we provide a brief introduction to the $(k, z)$-center problem and the dynamic model used in this paper.

Let $P$ be a set of data points. For any two points $x, y \in P$, denote $\delta(x, y)$ as their distance. We assume that the algorithm is given oracle access to distances in the underlying metric, where a query to $\delta(x, y)$ for any $x, y \in P$ can be answered in $O(1)$ time. Given a subset $C \subseteq P$, denote $\delta(p, C) = \min_{c \in C} \delta(p, c)$ as the distance from a point $p$ to its closest center in $C$. For any two sets $\mathcal{A}, \mathcal{B} \subseteq P$, let $\delta(\mathcal{A}, \mathcal{B}) = \max_{x \in \mathcal{A}} \delta(x, \mathcal{B})$ be the clustering cost of $\mathcal{A}$ induced on the set $\mathcal{B}$. Given two point sets $A$ and $C$ together with a radius $r \in \mathbb{R}$, define $B(A, C, r) = \{x \in C : \delta(x, A) \leq r\}$ as the subset of $C$ covered by balls centered at points in $A$ with radius $r$. For a $(k, z)$-center instance, we use $\mathcal{I}$ and $\mathcal{Z}^*$ to denote the sets of true inliers and outliers in the optimal solution (ties are broken arbitrarily), respectively. The $k$-center with outliers problem (also denoted as the $(k, z)$-center problem) is defined as follows.

**Definition 2.1.** Let $(P, k, z)$ be a given $(k, z)$-center instance in a metric space, where $P$ is the point set, $k \in \mathbb{N}$ is the number of clusters, and $z \in \mathbb{N}$ is the outlier budget. The goal of $(k, z)$-center problem is to find a set $C \subseteq P$ of centers with $|C| \leq k$ and a candidate outlier set $\mathcal{Z} \subseteq P$

---

[1]Note that in this paper, we use $\tilde{O}$ notation to hide polylogarithmic factors in the data size $n$, where the detailed time complexity with big O notations is given in the Appendix

with $|\mathcal{Z}| \leq z$ such that the maximum distance from points in $P \backslash \mathcal{Z}$ to their closest centers in $C$ is minimized, i.e., $\min_{C \subseteq P, \mathcal{Z} \subseteq P : |C| \leq k, |\mathcal{Z}| \leq z} \max_{p \in P \backslash \mathcal{Z}} \delta(p, C)$.

In the fully-dynamic setting, we also assume that the algorithm is given oracle access to distances in the underlying metric. The dataset $P$ evolves through a dynamic stream that starts from an initial point set $P$ (note that $P$ can be an empty set) and then receives a sequence of insertion and deletion operations. At each time step $t$, either a new point is inserted into the current dataset or an existing point is deleted, where deletions are allowed only for points that are currently present. We use $P_t$ to denote the point set after the $t$-th update. When it is clear from the context, the subscript is omitted. Given a positive integer $T$, we use $[T]$ to denote the set $\{1, 2, ..., T\}$. Let $d_{\min}^t$ and $d_{\max}^t$ denote the minimum and maximum pairwise distances that appear at time step $t$ during the update sequence, namely $d_{\min}^t = \min\{\delta(x, y) : x, y \in P_t\}$ and $d_{\max} = \max\{\delta(x, y) : x, y \in P_t\}$. Define the aspect ratio as $\Delta = \max_{t \geq 0} d_{\max}^t / d_{\min}^t$. Unlike previous work (Biabani et al., 2024; Chan et al., 2022), we assume no prior knowledge of $\Delta$ or any bounds on pairwise distances.

**The Fully-Dynamic $(k, z)$-center Problem.** In this problem, we are given a dynamic stream. At each time step $t$, the input consists of the current point set $P_{t-1}$ and parameters $k$ and $z$, where $k$ is the number of clusters and $z$ is the outlier budget. The goal is to maintain dynamic data structures with efficient update time such that, upon receiving any query at time step $t$, a center set $C \subseteq P_t$ can be returned with minimized $(k, z)$-center objective on the instance $(P_t, k, z)$.

**Roadmap.** The rest of the paper is organized as follows. Section 3 introduces our main approach: we construct a layered weighted summary that preserves enough information to recover a constant-factor approximation for $k$-center clustering with outliers. Section 4 further accelerates this framework by refining the update and query procedures, leading to faster dynamic performance. We provide the complete algorithmic descriptions, auxiliary data structures, and full proofs in the appendices, including the static construction, the dynamic invariants, the approximation analysis, and the detailed update-time and query-time bounds.

## 3. Aspect Ratio Independent Algorithms for Fully-Dynamic $(k, z)$-center Problem

In this section, we present new algorithms for the fully-dynamic $(k, z)$-center problem. The main objective is to design algorithms with efficient update and query time that are independent of the aspect ratio $\Delta$, while maintaining a constant-factor approximation. However, as pointed out in previous work (Chan et al., 2024), achieving any constant approximation with exactly $z$ outliers discarded requires

$\Omega(z)$ update time. In many practical scenarios, outliers may take a non-negligible fraction of the clustering data (i.e., $z = \Theta(n)$), where the lower bound update time becomes $\Omega(n)$. This can limit the efficiency and scalability of dynamic algorithms with exactly $z$ outliers discarded. Hence, to achieve sub-linear update time, the focus is on the bi-criteria settings that allow discarding up to $(1 + \epsilon)z$ outliers.

The intuitive idea behind our proposed method is to dynamically maintain small weighted summaries (similar to weak coreset or weighted instance) induced by a sequence of layered structures, where queries can be answered by executing an existing $(k, z)$-center routine on the summaries to obtain the final clustering solutions. To avoid the radius guessing, each layer consists of a random sampled representative set that covers certain fractions of the currently uncovered points, where the covering radius is determined adaptively by these representatives and the points they capture.

Although the sampling and greedy covering strategy avoids the radius guessing process, this process does not provide the desired guarantees on the number of outliers discarded (Chen et al., 2018). To address this issue, we propose a new sampling-based approach that separates the layered structure constructions into different stages. The initial layers cover most inliers and shrink the uncovered set geometrically. Once the uncovered set becomes comparable to the outlier budget, a transition layer is constructed by slightly increasing the sample size such that most large optimal clusters can be captured, leaving at most $(1 + \epsilon)z$ uncovered points. However, the uncovered residue may still contain more than $\epsilon z$ inliers in the worst case. Directly executing a $(k, z)$-center routine on the representatives can lead to far more than $(1 + \epsilon)z$ discarded outliers. Therefore, the final layers perform a fine-grained division of the residual points and incorporate them in small batches by updating only the weights of the transition representatives. This yields a sequence of weighted summaries that gradually recover the discarded inliers. Among them, there exists one summary with at most $\epsilon z$ uncovered inliers, which can yield a constant approximation while discarding $(1 + \epsilon)z$ outliers.

During the updates, the sampling distributions are maintained using Bernoulli processes. To guarantee sub-linear update time, the layered structures are maintained through an update-delay rule, where reconstruction is triggered only after insertions and deletions accumulate beyond a certain threshold. This spreads the rebuild cost over a sequence of updates, and we show that the delayed maintenance introduces only a constant-factor loss in approximation guarantees. For the query process, we propose a greedy preprocessing method that merges representatives close to each other in the weighted summary. After the merging step, the remaining representatives satisfy a well-separated property. Under this structure, either an approximate optimal cluster-

ing radius can be captured by the pairwise distances between representatives, or the top-$k$ representatives with the largest weights already cover almost all inliers and yield a feasible solution. In both cases, queries are answered without using any $\Delta$-dependent radius-search procedure.

The rest of this section presents the formal algorithms and analyses. We first propose the static layered-sampling method in the offline setting and describe data structures required. Then, we show that the offline setting can be extended to the fully-dynamic setting with slight modifications on the data structures by incorporating update-delay rules.

### 3.1. The Static Algorithm

Our starting point is a static algorithm which gives a $24(1 + \epsilon)$-approximation for the $(k, z)$-center problem. Due to space limit, we present only a high-level description here and leave the detailed analysis in appendix (Appendix B).

The high-level idea behind is to compress the input into a sequence of layered structure (denoted as data structure $\mathcal{D}$) consisting of randomly selected representatives. Instead of relying on a fixed-radius covering strategy induced by optimal-radius guessing, the proposed layered-sampling method covers a fixed fraction of the currently uncovered points in each iteration based on the sampled representatives, where the covering radius is adaptively determined by the distance to the farthest covered point. In general, the layered structure $\mathcal{D}$ should satisfy the following properties.

- The data structure $\mathcal{D}$ maintains a collection of $h$ layers. These layers are further divided into three types by a separating index $i^* < h$. Layers with indices $j < i^*$ form the type-1 layer, the layer with index $j = i^*$ forms the type-2 layer (also denoted as the transition layer), and layers with indices $j > i^*$ form the type-3 layer.

- Each layer is represented by a tuple $(\mathcal{U}_i, S_i, X_i, \rho_i, w_i)$ (for some index $i \leq h$), where $\mathcal{U}_i$ denotes the set of currently uncovered data points, $S_i$ is the set of representatives in the $i$-th layer, $X_i \subseteq \mathcal{U}_i$ denotes the set of points covered in the $i$-th layer by balls of radius $\rho_i$ centered at representatives in $S_i$, and $w_i : S_i \rightarrow \mathbb{R}$ is a mapping function that assigns a weight $w_i(p)$ to each $p \in S_i$.

- It is guaranteed that for each type of layer, the covered point set takes a certain fraction of the uncovered point set, i.e., $|X_i| = \alpha_i |\mathcal{U}_i|$ holds for some $0 < \alpha_i < 1$.

To construct the layered structure $\mathcal{D}$, the proposed layered-sampling algorithm (detailed in Algorithm 1) proceeds in a sequence of stages. At the very beginning, the full dataset is considered as uncovered points (step 1 of Algorithm 1). The initial layers cover a constant fraction of the uncovered

---

**Algorithm 1** Layered-Sampling$(P, k, z, \alpha, \epsilon)$

---

**Input:** A $k$-center with outliers instance $(P, k, z)$, parameters $0 < \alpha \leq 1/2, 0 < \epsilon < 1$.

**Output:** A set $\mathcal{C}$ of clustering centers.

1: $i \leftarrow 1, \mathcal{U}_1 \leftarrow P, \mathcal{C} \leftarrow \emptyset, \xi \leftarrow \max\{k, \log |P|\}, .$

2: **while** $|\mathcal{U}_i| > 8z$ **do**

3:     Sample a set $S_i$ of $\Theta(\xi)$ points randomly and independently from $\mathcal{U}_i$.

4:     Let $\rho_i$ be the smallest radius s.t. $|B(S_i, \mathcal{U}_i, \rho_i)| \geq \alpha |\mathcal{U}_i|$.

5:     $X_i \leftarrow B(S_i, \mathcal{U}_i, \rho_i), \mathcal{C} \leftarrow \mathcal{C} \cup S_i$.

6:     Assign each $p \in X_i$ to its nearest center $\sigma(p) = \arg\min_{s \in S_i} \delta(p, s)$, and set $w_i(s) = \big|\{p \in X_i : \sigma(p) = s\}\big|$ for each $s \in S_i$ .

7:     $\mathcal{U}_{i+1} \leftarrow \mathcal{U}_i \setminus X_i, i \leftarrow i + 1$.

8: **end while**

9: $i^* \leftarrow i$.

10: Call Residual-Peeling$(i^*, \mathcal{U}_{i^*}, k, z, \epsilon)$ algorithm to obtain the maximum layer index $h$, and a sequence of layered structures $(\mathcal{U}_j, S_j, X_j, \rho_j, w_j)$ for $j \in [i^*, h]$. {Detailed in Algorithm 3 in Appendix B}

11: Construct $\mathcal{D} = \{(\mathcal{U}_j, S_j, X_j, \rho_j, w_j) : j \in [1, h]\}$

12: $\mathcal{C}_f = $ Center-Selection$(\mathcal{D}, i^*, h, k, z, \epsilon)$. {Detailed in Algorithm 4 in Appendix B}

13: **return** $\mathcal{C}_f$.

---

points and aggregates them onto small sets of representatives (steps 2-8 of Algorithm 1). By repeating this procedure, the algorithm ensures that the vast majority of inliers can be covered with bounded radii, where only $O(z)$ points remain uncovered (the stopping condition in step 2 of Algorithm 1) after this stage. To handle the uncovered points, a residual peeling procedure (step 10 of Algorithm 1) is proposed, where the representative sample size is slightly increased to cover most inliers belonging to large optimal clusters. During the Residual-Peeling stage, a fine-grained division strategy is used to recover missed inliers (see Appendix B for details), where uncovered points are incorporated into the representatives in small batches. It can be guaranteed that there exists at least an incorporation at which the number of uncovered inliers is bounded roughly by $\epsilon z$.

The maintained representatives and their weights can be leveraged to construct weighted instances (also called weighted summaries). Then, any query can be answered by executing a $(k, z)$-center routine on the summaries to return the final clustering solution. However, a remaining issue is that existing $(k, z)$-center algorithms typically rely on radius guessing to maintain uniform-radius clusters, which may reintroduce aspect-ratio dependence in the query time. Thus, on top of the layered structures, our algorithm proposes to adapt a greedy center selection step to avoid the radius guessing process (step 12 of Algorithm 1). It first applies

a pre-precessing step (details in Appendix B) to merge the candidates close enough to each other, where the remaining representatives can behave well-separated properties. Then, there are two cases that may happen: (1) a binary search over the pairwise distances between the representatives can yield an approximate optimal radius such that a final center set can be constructed with provable guarantees; or (2) selecting the top-$k$ representatives with largest weights directly yields satisfied clustering solutions. In both cases, the $\Delta$-dependent radius guessing process can be avoided.

The following theorem shows that with constant probability, the proposed Layered-Sampling algorithm can yield a $24(1+\epsilon)$-approximate solution for the $(k, z)$-center problem with near-linear running time in the data size, where the number of outliers discarded can be bounded by $(1 + \epsilon)z$.

**Theorem 3.1.** *With constant probability, Layered-Sampling can return a $24(1 + \epsilon)$-approximate solution in time $\tilde{O}(nk/\epsilon^2 + k^2/\epsilon^4)$ with $(1+\epsilon)z$ outliers discarded.*

Note that the time complexity for the static algorithm (Algorithm 1) with big $O$ notation is $O(nk \log n + \frac{nk \log k}{\epsilon^2} + \frac{k^2 \log^2 n}{\epsilon^3} \log(\frac{k \log n}{\epsilon}) \log\left(\frac{k \log n}{\epsilon^2}\right))$, where the detailed analysis is presented in Appendix B.

### 3.2. The Dynamic Algorithm

The intuitive idea behind the dynamic algorithm is to maintain similar layered structure as in the static setting. To achieve efficient update time, an update-delay strategy is adapted to postpone the reconstructions until the insertions and deletions accumulate beyond a certain threshold.

In general, the layered structure $\mathcal{D}'$ in the dynamic setting follows a similar form as in the static setting. The difference is that each layer maintains a counter that records the number of insertions and deletions since its last reconstruction. Once the counter reaches a fixed fraction of the size for the current uncovered set, the structure $\mathcal{D}'$ is reconstructed. In addition, the structure $\mathcal{D}'$ maintains a set $\mathcal{Z}$ of candidate outliers, which collects points from insertions that are not updated immediately. The properties of the dynamic data structure $\mathcal{D}'$ is summarized as follows.

- The data structure $\mathcal{D}'$ maintains a collection of $h$ layers. These layers are further divided into three types by a separating index $i^* < h$. Layers with indices $j < i^*$ form the type-1 layer, the layer with index $j = i^*$ forms the type-2 layer (also denoted as the transition layer), and layers with indices $(j > i^*)$ form the type-3 layer.

- Each layer is represented as a tuple $(\mathcal{U}_i, S_i, X_i, \rho_i, w_i)$ (for some index $i \leq h$), where $\mathcal{U}_i$ denotes the set of currently uncovered data points, $S_i$ is the set of representatives in the $i$-th layer, $X_i \subseteq \mathcal{U}_i$ denotes the set of points

covered in the $i$-th layer by balls of radius $\rho_i$ centered at representatives in $S_i$, and $w_i : S_i \to \mathbb{R}$ is a mapping function that assigns a weight $w_i(p)$ to each $p \in S_i$.

- It is guaranteed that for each type of layer, the covered point set takes a certain fraction of the uncovered point set, i.e., $|X_i| = \alpha_i |\mathcal{U}_i|$ holds for some $0 < \alpha_i < 1$.

- Each type-1 and type-2 layer maintains a counter $n_i$, which denotes the number of updates to $\mathcal{U}_i$ since the last time $\mathcal{U}_i$ is rebuilt. For each layer with index $i < i^*$, let $\mathcal{U}_i^{\text{OLD}}$ denote the state of the uncovered set in the $i$-th layer since the last time it was reconstructed. The counter is used to trigger the updates of the layered structure. There is a parameter $0 < \tau < 1$ (to be specified) such that $n_i \leq \tau |\mathcal{U}_i^{\text{OLD}}|$ holds for each layer with index $i \in [1, i^*]$.

- The data structure $\mathcal{D}'$ also maintains a set $\mathcal{Z}$ of candidate outliers during the whole dynamic update processes.

The formal description for the proposed dynamic algorithm is presented in Algorithm 2. Due to space limit, we give a high-level description of the main update strategies, leaving detailed analysis in appendix.

---

**Algorithm 2** FastDynamicClustering$(P, k, z, \epsilon, \alpha, \tau)$

---

**Input:** A sequence $\mathcal{S} = \{o_1, o_2, ..., o_i\}$ of update operations, an initial dataset $P$, parameters $k, z, \epsilon, \alpha, \tau$.

1: Initialize a data structure $\mathcal{D}'$ by calling the Layered-Sampling$(P, k, z, \alpha, \epsilon)$ algorithm.
2: Initialize $n_i = 0$ for each $i \in [i^*]$, and set $\mathcal{Z} = \emptyset$, where $i^*$ is the separating index for $\mathcal{D}'$.
3: **for** $o_i \in \mathcal{S}$ **do**
4:     **if** $o_i$ is to insert a point $x$ **then**
5:         Call Insert$(x, \tau)$ to update $\mathcal{D}'$. {Detailed in Algorithm 6 in Appendix C}
6:     **else if** $o_i$ is to delete a point $x$ **then**
7:         Call Delete$(x, \tau)$ to update $\mathcal{D}'$. {Detailed in Algorithm 7 in Appendix C}
8:     **end if**
9:     For a specific query, call the Center-Selection Algorithm (Algorithm 4) based on $\mathcal{D}'$ to obtain a solution by setting the input $z$ as $z - |\mathcal{Z}|$, where $\mathcal{Z}$ is the set of candidate outliers maintained by $\mathcal{D}'$.
10: **end for**

---

**Handling the Insertion:** The algorithm for handling the insertion operations are presented in Algorithm 6 in Appendix C. After receiving an operation for inserting a point $x$, it is simply regarded as an uncovered point in type-1 layers, where the algorithm adds $x$ to the uncovered set $\mathcal{U}_i$ for each type-1 layer. Meanwhile, the counters (i.e., $n_i$) are also updated to record how many changes have been made since last rebuild. Then, we need to consider that $x$ might be

selected as a representative in the transition layer. To maintain the sampling distribution, a Bernoulli sampling step is performed on $x$ to simulate the event that $x$ is selected uniformly at random as the representatives in the transition layer. If $x$ is sampled as a representative, then the layered structure is rebuilt from the transition layer to maintain the insertion invariant. Otherwise, the algorithm treats $x$ as a candidate outlier and inserts it into the uncovered sets $\mathcal{U}_i$ for all $i \in [i^*, h]$ as well as the outlier candidate set $\mathcal{Z}$. Once the counters indicate that a certain fraction of updates (i.e., $n_i \geq \tau \cdot |\mathcal{U}_i^{\mathrm{OLD}}|$ for some parameter $0 < \tau < 1$) have been made for a specific layer, then the algorithm reconstructs the layered structures from such layer and reset the counter.

**Handling the Deletion:** The algorithm for handling the deletion operations are presented in Algorithm 7 in Appendix C. For each deletion operation that removes a point $x$ from the current data, there are several cases that may happen: (1) $x$ is a representative in $S_i$ for some layer with index $i < i^*$; (2) $x$ is a representative in the transition layer; (3) $x$ is a covered point in $X_i$ for some layer index $i < h$; (4) $x$ belongs to the set $\mathcal{Z}$ of candidate outliers. If case (1) happens, the point $x$ is removed from $S_i$, and an alternative $x' \in X_i$ with $x' \neq x$ and $\sigma(x) = \sigma(x')$ is selected to replace $x$ as a new representative. If case (2) happens, then the layers are rebuilt from the $i^*$-th layer (i.e., the transition layer) since the current representative set $S_{i^*}$ may no longer follow a uniform sampling distribution. Finally, if case (3) or case (4) happens, $x$ is removed directly from $X_i$ or $\mathcal{Z}$. Meanwhile, the algorithm updates the counters to record how many changes have been made since the last rebuild. Once the counters indicate that a certain fraction of updates (i.e., $n_i > \tau \cdot |\mathcal{U}_i^{\mathrm{OLD}}|$ for some parameter $\tau$) have been made for a specific layer, then the algorithm reconstructs the layered structures from such layer and reset the counters.

**The Rebuild Procedure:** The algorithm for rebuilding the layered structure is presented in Algorithm 8 in Appendix C. Given a layer index $j$, the rebuild procedure reconstructs the layered structure from the $j$-th layer by invoking the static algorithm with the uncovered set $\mathcal{U}_j$ as the input. Meanwhile, the counters and separating index are reset accordingly.

Based on the update strategies, the layered structure can maintain the following invariant during the updates.

**Invariant 3.2.** *Given a parameter $0 < \tau < 1$, it holds that $n_i \leq \tau |\mathcal{U}_i^{\mathrm{OLD}}|$ for each $i \in [1, i^*]$.*

Intuitively, the above invariant ensures that the layered structure maintained by our dynamic algorithm remains close to the layered structures of the static setting in Section 3.1. Based on this property, there are $\tilde{O}(1/\epsilon)$ layers maintained. During the updates, the event that an insertion or deletion affects the representative set in the transition layer occurs with uniform probability distribution according to Bernoulli sampling steps. Thus, the expected number of reconstructions

is small with update time independent of the data size.

**Lemma 3.3.** *Exclude the Rebuild procedure, the expected update time for our dynamic algorithm is $\tilde{O}(k^2/\epsilon^3)$.*

If the reconstruction is invoked when a specific layer counter reaches its threshold, then the rebuild cost can be amortized over the updates since the last reconstruction, where a sublinear time bound can still be obtained.

**Lemma 3.4.** *If a reconstruction is invoked when a specific layer counter reaches its threshold, then the amortized update time for reconstruction can be bounded by $\tilde{O}(k/\epsilon^4)$.*

For approximation guarantees, Invariant 3.2 intuitively indicates that the dynamic structure $\mathcal{D}'$ can share similar properties of the structure $\mathcal{D}$ in static settings. Upon receiving a specific query, by invoking a center selection algorithm as used in static settings (i.e., Algorithm 4 in Appendix B), an $O(1)$-approximate solution can be obtained with $(1 + \epsilon)z$ outliers discarded. Putting all these together, Theorem 1.1 can be proved (detailed proofs are in Appendix C).

# 4. Faster Algorithm for Fully-Dynamic $(k, z)$-center via Data Compression

In this section, to further speed up the update and query time, we propose sampling-based schemes to reduce the data size while maintaining approximation guarantees under mild assumption of optimal cluster sizes. Specifically, for the $(k, z)$-center problem, an independent random sampling scheme (Lemma 4.1) is available to construct a small subset of the original data with preserved theoretical guarantees.

**Lemma 4.1.** *Let $(P, k, z)$ be a $k$-center with outliers instance, and let $S \subseteq P$ be a set obtained by sampling each data point independently with probability $p = \frac{4\ln(2k)}{\epsilon z}$. Define a new instance $(S, k, \hat{z})$ with $\hat{z} = \frac{4\ln(2k)}{\epsilon \eta}$ for some parameter $0 < \eta < 1$ such that $\epsilon > \frac{2}{3\eta}$. Under the assumption that each optimal cluster in $P$ has size at least $3\epsilon z$, if an algorithm $\mathcal{A}$ returns a $\zeta$-approximate solution with $z$ outliers discarded, a $(\zeta + 2)$-approximation on $P$ with $z$ outliers discarded can be achieved with probability $\Omega(1 - \eta)$ by executing $\mathcal{A}$ on $S$ with $\hat{z} = \frac{4\ln(2k)}{\epsilon \eta}$.*

Based on the data compression scheme, our strategy is to adapt Bernoulli sampling steps to simulate the randomly independent sampling distribution during the updates, where the update and query time can be further improved by a factor of $z$ (detailed proofs are in Appendix D). For each insertion, a Bernoulli sampling step (step 5 of Algorithm 10) is performed to simulate that $x$ is sampled to form the weak coreset $S$. Since the sampling probability is independent of the data size, the insertion and deletion operations on the points are independent of each other, while guaranteeing the probability distributions for constructing $S$.

We further complement this result with a lower bound showing that, in the general metric space query model, any algorithm with $O(1)$-approximation and $O(z)$ outliers discarded must incur $\Omega(k^2/z)$ update time (see Appendix D). At a high level, the theorem is proved by constructing a distribution over instances in the general metric space query model such that no deterministic algorithm can succeed with probability exceeding $0.5$ under this distribution. Hence, the stated randomized lower bound then follows by Yao's minimax principle (Yao, 1977).

To prove a lower bound for the update time, our strategy is to transfer this static lower bound to the dynamic setting. A key observation is that the query time has lower bound of $\Omega(k)$ since there are $k$ clusters. Consider an incremental $k$-center data structure with amortized update time $u(n, k, z)$ and query time $q(n, k, z)$. By inserting the $n$ points of one at a time and then raise a query when all the points have been inserted, the dynamic algorithm can be modified in this way to obtain an approximate solution in static settings with an overall running time of $n \cdot u(n, k, z) + q(n, k, z)$. According to Theorem D.1 (see Appendix for details), we have $n \cdot u(n, k, z) + q(n, k, z) = \Omega(nk^2/z)$ in the the general metric space query model. Then, since a fully-dynamic $(k, z)$-center algorithm should have $\Omega(k)$ query time, an update time lower bound of $\Omega(k^2/z)$ can be obtained

## 5. Conclusions and Discussions

**Conclusions.** This paper presents constant-factor approximation algorithms for the fully dynamic $(k, z)$-center problem that achieve $\Delta$-independent update and query times with $(1 + \epsilon)z$ discarded outliers. To the best of our knowledge, this is the first fully dynamic $(k, z)$-center approach that removes aspect-ratio dependence with provable guarantees. An interesting future direction is to extend our proposed methods to MPC settings. Our method is built from two main ingredients: independent sampling and ball division. The sampling component is based on uniform sampling and is naturally parallelizable, while the main obstacle is the ball-division step. In geometric settings, a possible route is to combine with hashing-based tools such as consistent hashing or LSH to approximately implement this step. Additionally, another interesting future direction is to extend the proposed framework to k-median/k-means objectives, although extending it would require new analysis.

**Discussion of Limitations.** A potential limitation of our proposed methods is that the improvements in the update and query times depend on the ranges of the aspect ratio for the given clustering instances. When the aspect ratio is polynomially bounded in the data size, the $\log \Delta$ dependence in prior work can be comparable to the $O(\log n)$ and additional $O(\log k)$ factors in our bounds. However, as

pointed out in prior work (Cohen-Addad et al., 2022), $\Delta$ can be much larger than polynomial in the input size, and in the worst case may even be arbitrarily large (Bhattacharjee & Moshkovitz, 2021). Such large aspect ratios can also arise in practice, for example in high-precision sensing, or datasets with many near-duplicate points, and this can naturally occur in fully dynamic settings. In these regimes, removing the explicit dependence on $\Delta$ yields a clear advantage. Additionally, for our proposed algorithms, the removal of aspect ratio can induce larger approximation loss in clustering quality guarantees compared with previous results that are dependent on the aspect ratio $\Delta$. How to develop new algorithms with improved approximation guarantees is an interesting problem that deserves further study.

**Broader Practical Implications.** The $k$-center problem belongs to the same family of center-based clustering objectives as $k$-median and $k$-means, differing mainly in how assignment costs are defined. Thus, $k$-center captures a complementary notion of clustering quality, which is also relevant in ML. Additionally, both $k$-center and $k$-center with outliers have been extensively studied in the ML community, including coreset-based batch active learning and settings with noisy or fully dynamic data. In such regimes, worst-case coverage and robustness to outliers are natural requirements, making fully dynamic $k$-center with outliers relevant to ML rather than purely algorithmic. Methodologically, our contribution is not only a new result for $k$-center, but also an extension of layered sampling to robust fully dynamic setting. More broadly, it can serve as a useful tool for practical ML algorithms on large-scale and noisy data.

## Acknowledgements

This work was supported by National Natural Science Foundation of China (62502545, 62432016), the Science and Technology Innovation Program of Hunan Province (2025RC3207), Open Project of Xiangjiang Laboratory (25XJ02007). This work was also carried out in part using computing resources at the High Performance Computing Center of Central South University.

## Impact Statement

This paper develops more efficient algorithms for clustering with outliers in fully-dynamic settings, where the update and query times can be independent of the aspect ratio of the clustering instances. The proposed method can help to reduce computation costs and make large-scale clustering more accessible. The contributions are purely algorithmic and theoretical, and we do not anticipate negative societal impacts. Our study does not involve human subjects, or identifiable information, and no specific ethical concerns arise beyond these general considerations.

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

## Technical Appendices and Supplementary Material

## A. Introduction to the Existing Algorithms for Static $(k, z)$-center Problem

In the following, we summarize the approximation results for the $(k, z)$-center problem in static settings.

The $k$-center with outliers problem (i.e., the $(k, z)$-center problem) was formulated by Charikar et al. (2001), where a set of $z$ data points is allowed to be discarded as outliers while minimizing the standard $k$-center objective. Following this formulation, Charikar et al. (2001) proposed a polynomial-time 3-approximation algorithm in metric spaces using a greedy coverage strategy. Chakrabarty & Negahbani (2019) proposed a reduction-based approach, achieving an improved 2-approximation in polynomial time, where the approximation guarantees match the approximation lower bound on clustering quality for the $k$-center objective (Gonzalez, 1985). Moreover, Grunau & Rozhoň (2022) demonstrated that achieving any constant approximation for the $(k, z)$-center problem with exactly $z$ outliers discarded requires a time complexity of $\Omega(z^2)$. In the worst case when the dataset contains significant noise (i.e., $z = \Omega(n)$), the runtime lower bound becomes quadratic.

To obtain near-linear (or linear) running time in the data size, a separate line of research focuses on bi-criteria approximation algorithms, where the number of clusters or the outlier budget is allowed to be slightly relaxed. In this setting, Ding et al. (2019) developed a sampling-based algorithm that achieves a 2-approximation in time $O(nk/\epsilon)$, where the number of centers opened is $O(k/\epsilon)$ and the number of outliers discarded is $(1 + \epsilon)z$. Bhaskara et al. (2019) proposed a greedy method achieving a $(2 + \epsilon)$-approximate solution in time $O(nk \log \log(n\Delta)/\epsilon)$, while discarding $O(z \log k)$ outliers. Grunau and Rozhoň (Grunau & Rozhoň, 2022) proved a runtime lower bound of $O(nk^2/z)$ for achieving any $O(1)$-approximation with $O(z)$ outliers discarded. In distributed settings, several constant approximations with $(1 + \epsilon)z$ outliers discarded were known (Li & Guo, 2018; Huang et al., 2023). However, these methods still incur dependencies on the aspect ratio terms, either in the communication cost (Li & Guo, 2018) or in the coordinator's running time (Huang et al., 2023).

## B. The Static $(k, z)$-center Algorithm

In this section, we provide a detailed analysis for the proposed static $(k, z)$-center algorithm (i.e., the Layered-Sampling algorithm in Section 3.1). At a high level, the proposed algorithm constructs a sequence of weighted instances (also called the weighted summaries) of size $\tilde{O}(k/\epsilon)$ by selecting representative from the dataset. Each representative is associated with an integer weight denoting the number of points in the given dataset that are assigned to it. These weighted representatives help to compress the data sizes, while maintaining provable clustering-quality guarantees.

As discussed in the main context, a recurring issue in existing weighted summary construction is that the summary size should usually depend either on the aspect ratio $\Delta$ (Chan et al., 2022; Biabani et al., 2024; Li & Guo, 2018) or on the outlier budget $z$ (Chen et al., 2018; Huang et al., 2025; Deshpande et al., 2020). To address this issue, we propose to construct the representatives using a layered structure, where different types of layers are maintained for different roles. Starting from the full dataset as the uncovered set, the layered structure is constructed by covering a fixed fraction of the currently uncovered points in each layer. During the covering process, the layers with smaller indices select representatives that can cover the majority of inliers, where the number of uncovered inliers shrinks geometrically and only a small fraction of inliers remain outside these layers (with roughly $O(z)$ uncovered points). The layers with larger indices are constructed on the remaining uncovered points, aiming to recover inliers that are missed by the earlier layers. Based on these layered structures, the algorithm forms summaries by combining different layer types, producing a sequence of candidate summaries associated with different outlier budgets. Among the constructed summaries, we will show that there exists one summary in which only $\epsilon z$ inliers are discarded. Finally, by executing a weighted $(k, z)$-center routine (i.e., the 3-approximation greedy algorithm (Charikar et al., 2001)) on the summaries, an $O(1)$-approximate solution with $(1 + \epsilon)z$ outliers discarded can be obtained.

### B.1. The Data Structure Required

In the following, for better consistency, we also list the properties for the data structure required as presented in the main context. In general, the data structure $\mathcal{D}$ should satisfy the following properties.

- The data structure $\mathcal{D}$ maintains a collection of $h$ layers. These layers are further divided into three types by a separating index $i^* < h$. Layers with indices $j < i^*$ form the type-1 layer, the layer with index $j = i^*$ forms the type-2 layer (also denoted as the transition layer), and layers with indices $j > i^*$ form the type-3 layer.

- Each layer is represented by a tuple $(\mathcal{U}_i, S_i, X_i, \rho_i, w_i)$ (for some index $i \le h$), where $\mathcal{U}_i$ denotes the set of currently

uncovered data points, $S_i$ is the set of representatives in the $i$-th layer, $X_i \subseteq \mathcal{U}_i$ denotes the set of points covered in the $i$-th layer by balls of radius $\rho_i$ centered at representatives in $S_i$, and $w_i : S_i \to \mathbb{R}$ is a mapping function that assigns a weight $w_i(p)$ to each $p \in S_i$.

- It is guaranteed that for each type of layer, the covered point set takes a certain fraction of the uncovered point set, i.e., $|X_i| = \alpha_i |\mathcal{U}_i|$ holds for some $0 < \alpha_i < 1$.

The layered structure $\mathcal{D}$ is constructed to separate the instance into an "inlier part" and a small uncovered residue that contains most outliers together with a small fraction of inliers (the different parts are distinguished with the separating index $i^*$). More specific, layers with indices smaller than $i^*$ are constructed to select representatives that cover the majority of inliers. The transition layer with index $i^*$ is then constructed on this remaining uncovered set. Its role is to select additional representatives that can cover most of the missed inliers, with a covering radius related to the optimal clustering radius. Layers with indices larger than $i^*$ are used to progressively incorporate points from the uncovered residue by updating the weights of the representatives in the transition layer. The incorporation process proceeds in batches, where each such layer (with index larger than $i^*$) takes a small fraction of uncovered points that are closest to the representatives in $S_{i^*}$, while updating only the corresponding weights. This mechanism gradually brings back uncovered inliers, while the outliers that are far from all representatives remain excluded. It can be guaranteed that there exists one layer (with index larger than $i^*$) at which almost all inliers are incorporated whereas outliers far from the representatives are not included.

## B.2. Detailed Algorithmic Descriptions and Analyses

In this subsection, we present the static $(k, z)$-center algorithm together with its analysis, describing how the layered structure is constructed and how it yields the claimed approximation bounds.

Intuitively, our proposed static algorithm mainly consists of three stages: (1) Greedy-Coverage (steps 2-8 of Algorithm 1); (2) Residual-Peeling (step 10 of Algorithm 1, which is detailed in Algorithm 3); (3) Center-Selection (step 12 of Algorithm 1, which is detailed in Algorithm 4).

### B.2.1. THE GREEDY-COVERAGE STAGE

During the Greedy-Coverage stage, the goal is to cover most inliers by repeatedly selecting representatives that capture a fixed fraction of the currently uncovered points, thereby forming the type-1 layered structures. Starting from the full dataset, the algorithm iteratively covers a certain fraction of data points and maintains a set $\mathcal{U}_i$ (with $\mathcal{U}_1 = P$ as stated in step 1 of Algorithm 1) of uncovered data points. In each iteration, the algorithm samples a small set $S_i$ randomly and independently from $\mathcal{U}_i$ (step 3 of Algorithm 1) to serve as representatives. Note that for type-1 layers, the sample size (related to $\xi = \max\{k, \log n\}$ as stated in step 1 of Algorithm 1) is fixed to be sub-linear in the data size. We will show that this suffices to induce constant approximations. Then, based on the sampled representatives, the algorithm constructs balls centered at these points and chooses the smallest radius $\rho_i$ such that a constant fraction $\alpha$ of $\mathcal{U}_i$ can be covered (where $X_i$ is the resulting covered set, see steps 4–5 of Algorithm 1). The points in $X_i$ are then assigned to their nearest representatives in $S_i$, and the weight of a representative is defined as the number of points assigned to it (step 6 of Algorithm 1).

The Greedy-Coverage stage repeats this procedure as long as the number of uncovered points is still much larger than the outlier budget (i.e., $|\mathcal{U}_i| \geq 8z$ in step 2 of Algorithm 1 stated as the stopping condition). Since the procedure removes at least an $\alpha$-fraction of the current uncovered points in each round, the size of $\mathcal{U}_i$ shrinks geometrically and the total number of type-1 layers can be bounded by $O(\log n)$. At the end of this stage, most inliers have been covered by the sampled representatives, where only a relatively small fraction of inliers remain uncovered.

As shown in the literature (Chen et al., 2018), the Greedy-Coverage stage takes $O(nk \log n)$ time to construct $O(\log n)$ layers. For a specific layer index $i$, given the set $\mathcal{U}_i$ of uncovered points and any subset $S' \subseteq \mathcal{U}_i$ with size $\xi' = \lambda \cdot \max\{k, \log n\}$ for a large enough constant $\lambda$, we define $\rho(S', \mathcal{U}_i) = \min\{r > 0 : |B(S', \mathcal{U}_i, r)| \geq \alpha |\mathcal{U}_i|\}$ as the minimum radius such that balls centered at points in $S'$ can cover at least an $\alpha$-fraction of points in $\mathcal{U}_i$. Denote $\rho'_i = \min_{S' \subset \mathcal{U}_i : |S'| = \xi'} \rho(S', \mathcal{U}_i)$ as the optimal covering radius in the $i$-th layer. The following lemma shows that, with high probability, the type-1 layers constructed can induce coverage radii that are comparable to the optimal ones.

**Lemma B.1.** (Chen et al., 2018) *With probability at least $1 - 1/n^2$, we have $\rho'_i \geq \rho_i/2$ for all layer index $1 \leq i < i^*$.*

Let $L^*$ be the optimal clustering radius for a given $(k, z)$-center instance $(P, k, z)$. Based on Lemma B.1, we will establish a clear relationship between the optimal coverage radius $\rho'_i$ and the optimal clustering radius $L^*$. Recall that $\mathcal{I}$ is the set of

true inliers in the optimal solution for a given $(k, z)$-center instance $(P, k, z)$. Let $\mathcal{H}(P) = \{P_1^*, P_2^*, ..., P_k^*\}$ be the set of optimal clusters. The following lemma shows that $\rho_i'$ can always be bounded by a constant factor of the optimal radius if the number of uncovered points is larger than $8z$.

**Lemma B.2.** *For each $i \in [1, i^*)$, it holds that $\rho_i' \leq 2L^*$.*

*Proof.* Note that for a specific layer index $i < i^*$, we have $|\mathcal{U}_i| \geq 8z$ according to the stopping condition for the Greedy-Coverage stage. Denote $\mathcal{I}_i = \mathcal{I} \cap \mathcal{U}_i$ as the set of uncovered true inliers before constructing the $i$-th layer. Let $\mathcal{P}(\mathcal{U}_i) = \{P_h^* : P_h^* \in \mathcal{H}(P), P_h^* \cap \mathcal{U}_i \neq \emptyset\}$ be the set of optimal clusters that intersect with the uncovered points in $\mathcal{U}_i$. We will divide the optimal clusters of $\mathcal{P}(\mathcal{U}_i)$ into two different groups based on their sizes.

Define $G_i = \{P_h^* \in \mathcal{P}(\mathcal{U}_i) : |P_h^* \cap \mathcal{U}_i| \geq 3|\mathcal{U}_i|/(8|\mathcal{P}(\mathcal{U}_i)|)\}$ as the set of optimal clusters in $\mathcal{P}(\mathcal{U}_i)$ containing data points that account for at least a $3/(8|\mathcal{P}(\mathcal{U}_i)|)$ fraction of the uncovered points in $\mathcal{U}_i$. Let $M_i = \mathcal{P}(\mathcal{U}_i)\backslash G_i$ be the set of small optimal clusters, accordingly. Since $|\mathcal{U}_i| \geq 8z$ and there are only $z$ true outliers, we have $|\mathcal{I}_i|/|\mathcal{U}_i| \geq 7/8$. Then, it holds trivially that $|G_i| > 0$. Otherwise, the uncovered inliers only take strictly smaller than a $3/8$ fraction of the uncovered points, which contradicts with the fact that $|\mathcal{I}_i|/|\mathcal{U}_i| \geq 7/8$.

Next, consider an arbitrary optimal cluster $P_h^* \in G_i$. Let $c_h \in P_h^* \cap \mathcal{U}_i$ be an arbitrary uncovered data point in $P_h^*$. According to the triangle inequality, it holds trivially that the ball centered at $c_h$ with radius $2L^*$ (i.e., $B(c_h, \mathcal{U}_i, 2L^*)$) can cover all the data points in $P_h^* \cap \mathcal{U}_i$. Then, define $S_i'$ as the set of centers which contain exactly one data point from each $P_h^*$ such that $P_h^* \in G_i$. It holds that $|B(S_i', \mathcal{U}_i, 2L^*)| \geq (1 - 3/8 - 1/8)|\mathcal{U}_i| = 0.5|\mathcal{U}_i| \geq \alpha|\mathcal{U}_i|$, since we restrict that $0 < \alpha \leq 1/2$. This implies that $\rho_i' \leq 2L^*$, because $\rho_i'$ is the smallest radius over all center sets with size $\xi'$ ($\xi' > |S_i'|$) such that balls of radius $\rho_i'$ can cover at least an $\alpha$-fraction of uncovered points in $\mathcal{U}_i$. $\square$

By combining Lemma B.1 with Lemma B.2, we can bound the coverage radius of our algorithm for type-1 layers constructed.

**Corollary B.3.** *With probability at least $1 - 1/n^2$, we have $\rho_i \leq 4L^*$ for each $i \in [1, i^*)$.*

### B.2.2. THE RESIDUAL-PEELING STAGE

After the Greedy-Coverage stage, the separating index $i^*$ is fixed accordingly (see step 9 of Algorithm 1), while the number of uncovered points becomes comparable to the outlier budget $z$. Then, the algorithm enters the Residual-Peeling stage (step 10 of Algorithm 1, where the Residual-Peeling algorithm is detailed in Algorithm 3). The primary aim within this stage is to further reduce the number of uncovered inliers to roughly $\epsilon z$, where $\epsilon$ is the parameter for controlling the number of additionally discarded outliers. The procedure begins by drawing a larger sample size of representatives from the current uncovered set as new representatives (step 1 of Algorithm 3). Based on these representatives, the algorithm removes the $(1 + \epsilon)z$ points that lie farthest from $S_{i^*}$, forming an initial candidate outlier set $\mathcal{F}_{i^*}$ of outliers (step 2 of Algorithm 3). By assigning each data point in $\mathcal{U}_{i^*}\backslash\mathcal{F}_{i^*}$ to its nearest representative in $S_{i^*}$, the weights of representatives are updated accordingly (step 3 of Algorithm 3). To this end, the tuple $(\mathcal{U}_{i^*}, S_{i^*}, X_{i^*}, \rho_{i^*}, w_{i^*})$ forms the transition layer.

Following this initialization, the Residual-Peeling stage repeatedly processes the uncovered set $\mathcal{U}_i$ (for each layer with index $i > i^*$) by recovering the points that are closest to the representatives in $S_{i^*}$ (steps 5-10 of Algorithm 3). Specifically, in each iteration, the procedure takes a small batch $\mathcal{N}_i$ containing the nearest $\epsilon z/2$ points from the current uncovered set to the representatives in $S_{i^*}$ (step 6 of Algorithm 3), and incorporates them into the representatives in $S_{i^*}$ with updated weights (step 8 of Algorithm 3). These points are also accumulated into a covered set $\mathcal{R}$ (step 7 of Algorithm 3). To update the weights for the representatives, the covered points in the set $\mathcal{R}$ are assigned to their nearest representatives, and the corresponding weights are updated (step 8 of Algorithm 3). This procedure is repeated until all points are covered, which results in a sequence of type-3 layers.

In the following, we will show that the transition layer and the residual layers with indices larger than $i^*$ can yield good approximations on clustering quality. In addition, there exists an index $i \geq i^*$ such that the representatives can cover all but at most $\epsilon z$ inliers within a bounded radius, where the remaining uncovered inliers come only from small optimal clusters.

Recall that the Residual-Peeling stage first samples a set $S_{i^*}$ of size $\Theta(k \log k/\epsilon)$ as new representatives. The goal is to ensure that, after the sampling step, only a small number of inliers in $\mathcal{U}_{i^*}$ have distances larger than $2L^*$ to the representatives in $S_{i^*}$. Our analysis starts by dividing the remaining uncovered optimal clusters in the $i^*$-th layer into different groups. Let $\mathcal{P}(\mathcal{U}_{i^*}) = \{P_h^* : P_h^* \in \mathcal{H}(P), P_h^* \cap \mathcal{U}_{i^*} \neq \emptyset\}$ be the set of optimal clusters that intersect with the uncovered points in $\mathcal{U}_{i^*}$. Define $G_{i^*} = \{P_h^* \in \mathcal{P}(\mathcal{U}_{i^*}) : |P_h^* \cap \mathcal{U}_{i^*}| \geq \frac{\epsilon z}{2k}\}$ and $M_{i^*} = \mathcal{P}(\mathcal{U}_{i^*})\backslash G_{i^*}$ as the set of large and small optimal clusters,

---

**Algorithm 3** Residual-Peeling($i^*, \mathcal{U}_{i^*}, k, z, \epsilon$)

---

**Input:** A separating layer index $i^*$, a set $\mathcal{U}_{i^*}$ of the remaining uncovered data points, parameter $k$, $z$, $\epsilon$.

**Output:** A sequence of layered structures $\{(\mathcal{U}_i, S_i, X_i, \rho_i, w_i) : i \in [i^*, h]\}$, where $h$ is the maximum index of the layered structure.

1: Sample a set $S_{i^*}$ of size $\Theta(\frac{k \log k}{\epsilon})$ randomly and independently from $\mathcal{U}_{i^*}$.
2: Let $\mathcal{F}_{i^*}$ be the farthest $(1 + \epsilon)z$ points in $\mathcal{U}_{i^*}$ to $S_{i^*}$, and set $X_{i^*} = \mathcal{U}_{i^*} \setminus \mathcal{F}_{i^*}$.
3: Assign each $p \in X_{i^*}$ to its nearest center $\sigma(p) = \arg\min_{s \in S_{i^*}} \delta(p, s)$, set $\rho_{i^*} = \max_{x \in X_{i^*}} \delta(x, S_{i^*})$ and $w_{i^*}(s) = \big|\{p \in X_{i^*} : \sigma(p) = s\}\big|$ for each $s \in S_{i^*}$ .
4: $\mathcal{U}_{i^*+1} = \mathcal{U}_{i^*} \setminus X_{i^*}, i = i^* + 1, \mathcal{R} = \emptyset$.
5: **while** $|\mathcal{U}_i| > 0$ **do**
6:    Set $S_i = S_{i^*}$, and let $\mathcal{N}_i$ be the set of the nearest $\frac{\epsilon z}{2}$ points in $\mathcal{U}_i$ to $S_i$.
7:    $\mathcal{R} \leftarrow \mathcal{R} \cup \mathcal{N}_i, X_i \leftarrow \mathcal{N}_i$.
8:    Assign each $p \in X_i$ to its nearest center $\sigma(p) = \arg\min_{s \in S_i} \delta(p, s)$, and set $w_i(s) = w_{i^*}(s) + \big|\{p \in \mathcal{R} : \sigma(p) = s\}\big|$ for each $s \in S_i$ .
9:    $\mathcal{U}_{i+1} \leftarrow \mathcal{U}_i \setminus \mathcal{N}_i, i \leftarrow i + 1$.
10: **end while**
11: Set $h = i - 1$, and $\rho_i = \max_{p \in X_i} \delta(p, S_i)$ for each $i > i^*$.
12: **return** $\{(\mathcal{U}_i, S_i, X_i, \rho_i, w_i) : i \in [i^*, h]\}$.

---

respectively. The primary objective here is to sample at least one data point from each uncovered large optimal cluster such that all the large optimal clusters can be covered by the sampled representatives with radius $2L^*$. The following lemma shows that sampling $\Theta(k \log k / \epsilon)$ points suffices to provide a good approximation with constant probability.

**Lemma B.4.** *By sampling a set $S_{i^*}$ of size $\Theta(k \log k / \epsilon)$ randomly and independently from the set $\mathcal{U}_{i^*}$ in the $i^*$-th layer, with constant probability, $S_{i^*}$ contains at least one data point from each large optimal cluster $P_h^* \in G_{i^*}$.*

*Proof.* We first consider an arbitrary large optimal cluster $P_h^* \in G_{i^*}$. Define $\zeta = \frac{|P_h^* \cap \mathcal{U}_{i^*}|}{|\mathcal{U}_{i^*}|}$ as the fraction of points in $\mathcal{U}_{i^*}$ that belong to $P_h^*$. According to the definition of large optimal clusters, we have $|P_h^* \cap \mathcal{U}_{i^*}| \geq \frac{\epsilon z}{2k}$. This implies that $\zeta \geq \frac{\epsilon}{16k}$ since $|\mathcal{U}_{i^*}| < 8z$. When sampling a set $S_{i^*}$ uniformly and independently from $\mathcal{U}_{i^*}$, the probability that $S_{i^*}$ does not contain any data point from $|P_h^* \cap \mathcal{U}_{i^*}|$ is at most $(1 - \zeta)^{|S_{i^*}|}$. Hence, the success probability can be bounded by

$$\Pr[S_{i^*} \cap (P_h^* \cap \mathcal{U}_{i^*}) \neq \emptyset] \geq 1 - (1 - \zeta)^{|S_{i^*}|}.$$

To guarantee that the success probability is at least $1 - \eta$ for some constant $0 < \eta < 1$, it suffices to require that $1 - (1 - \zeta)^{|S_{i^*}|} \geq 1 - \eta$ (i.e., $(1 - \zeta)^{|S_{i^*}|} \leq \eta$). This implies that we only need to guarantee $|S_{i^*}| \geq \frac{\log(1/\eta)}{\log(1/(1-\zeta))}$, where an upper bound of $\frac{1}{\zeta} \log \frac{1}{\eta}$ can be established for $\frac{\log(1/\eta)}{\log(1/(1-\zeta))}$. Since $\zeta \geq \frac{\epsilon}{16k}$, it is sufficient to guarantee that $|S_{i^*}| \geq \frac{16k}{\epsilon} \log \frac{1}{\eta}$. By applying a union bound over all uncovered large optimal clusters and replacing $\eta$ with $\eta/k$, the sample size becomes $\Theta(k \log k / \epsilon)$. Then, $S_{i^*}$ contains one point from each large optimal cluster $P_h^* \in G_{i^*}$ with constant probability. $\square$

Lemma B.4 implies that if we take a set $S_{i^*}$ of representatives with size $\Theta(k \log k / \epsilon)$ randomly and independently from $\mathcal{U}_{i^*}$, then with constant probability, the set $S_{i^*}$ contains at least one point from each large optimal cluster $P_h^* \in G_{i^*}$. By using triangle inequality, the balls centered at $S_{i^*}$ with radius $2L^*$ can cover all the large optimal clusters in $G_{i^*}$. Hence, the total number of uncovered inliers can be bounded by $\epsilon z/2$. Let $\mathcal{I}_{i^*} = \mathcal{I} \cap \mathcal{U}_{i^*}$ be the set of true inliers in $\mathcal{U}_{i^*}$. Denote $\mathcal{U}(\mathcal{I}_{i^*}) = \bigcup_{P_h^* \in M_{i^*}} P_h^* \cap \mathcal{U}_{i^*}$ as the union of uncovered data points belonging to small optimal clusters. We can get that $\delta(\mathcal{U}_{i^*} \setminus \mathcal{F}_{i^*}, S_{i^*}) \leq \delta(\mathcal{U}_i^* \setminus (\mathcal{Z}^* \cup \mathcal{U}(\mathcal{I}_{i^*})), S_{i^*}) \leq 2L^*$, where $\mathcal{F}_{i^*}$ is the set of the furthest $(1 + \epsilon)z$ data points in $\mathcal{U}_{i^*}$ to $S_{i^*}$.

**Corollary B.5.** *With constant probability, it holds that $\delta(\mathcal{U}_{i^*} \setminus \mathcal{F}_{i^*}, S_{i^*}) \leq 2L^*$.*

However, in the worst case, the discarded $(1 + \epsilon)z$ data points (i.e., the set $\mathcal{F}_{i^*}$) may all belong to true inliers. This may lead to far more than $(1 + \epsilon)z$ outliers to be discarded when applying a weighted algorithm on the representatives to find exactly $k$ centers (with $(1 + \epsilon)z$ points remaining uncovered and up to $z$ additional points discarded as outliers by the weighted clustering routines). Consequently, once the centers are fixed (i.e., $S_{i^*}$), starting from step 5 of the Residual-Peeling algorithm, it repeatedly extracts the points in the current uncovered set $\mathcal{U}_i$ ($i > i^*$) that lie closest to $S_{i^*}$. Specifically, the

nearest $\epsilon z/2$ points from $\mathcal{U}_i$ to $S_{i^*}$ are selected as the batch of points to be incorporated in each iteration $i$ ($i > i^*$). Since each incorporated batch contains a very small fraction of points (i.e., only $\epsilon z/2$ points), we can get that there must exist at least one iteration (or a corresponding layer with index $i' \geq i^*$) such that there are at most $\epsilon z$ uncovered inliers (with $\epsilon z/2$ points belonging to uncovered small optimal clusters and $\epsilon z/2$ points failed to be recovered) in the set $\mathcal{U}_{i'+1}$, while all points in $P \backslash \mathcal{U}_{i'+1}$ have distances at most $4L^*$ to the sampled representatives. Let $\mathcal{V} = \bigcup_{i \in [1,i^*]} S_i$ be the union of the representatives, we have the following result.

**Corollary B.6.** *With constant probability, there exists a layer index $i' \in [i^*, h]$ such that $\delta(P \setminus \mathcal{U}_{i'+1}, \mathcal{V}) \leq 4L^*$ and $|\mathcal{U}_{i'+1} \cap \mathcal{I}| \leq \epsilon z$, where $\mathcal{V} = \bigcup_{j \in [1,i^*]} S_j$.*

*Proof.* Since Corollary B.3 holds with high probability and Lemma B.4 holds with constant probability, their intersection also holds with constant probability.

Let
$$\mathcal{I}_{\mathrm{sm}} = \bigcup_{P_h^* \in M_{i^*}} \left( P_h^* \cap \mathcal{U}_{i^*} \right)$$
be the set of uncovered inliers in small optimal clusters at the beginning of the Residual-Peeling stage, and let
$$\mathcal{I}_{\mathrm{lg}} = (\mathcal{I} \cap \mathcal{U}_{i^*}) \setminus \mathcal{I}_{\mathrm{sm}}$$
be the remaining uncovered inliers, which belong to large optimal clusters. By the definition of $M_{i^*}$, we have
$$|\mathcal{I}_{\mathrm{sm}}| \leq \sum_{P_h^* \in M_{i^*}} |P_h^* \cap \mathcal{U}_{i^*}| < k \cdot \frac{\epsilon z}{2k} = \frac{\epsilon z}{2}.$$

On the event of Lemma B.4, the sample $S_{i^*}$ contains at least one point from each large optimal cluster. Hence, by the triangle inequality, every point $p \in \mathcal{I}_{\mathrm{lg}}$ satisfies
$$\delta(p, S_{i^*}) \leq 2L^*.$$

We first note that the transition layer covers only points within distance $2L^*$ from $S_{i^*}$. Indeed, any point in $\mathcal{U}_{i^*}$ whose distance from $S_{i^*}$ is larger than $2L^*$ is either a true outlier or an inlier in $\mathcal{I}_{\mathrm{sm}}$. Therefore the number of such points is at most
$$z + |\mathcal{I}_{\mathrm{sm}}| \leq z + \frac{\epsilon z}{2} \leq (1 + \epsilon)z.$$

Since $\mathcal{F}_{i^*}$ consists of the farthest $(1 + \epsilon)z$ points in $\mathcal{U}_{i^*}$ from $S_{i^*}$ and $X_{i^*} = \mathcal{U}_{i^*} \setminus \mathcal{F}_{i^*}$, it follows that
$$\delta(X_{i^*}, S_{i^*}) \leq 2L^*.$$

Let $b = \epsilon z/2$ be the batch size used in the residual layers. If
$$|\mathcal{I}_{\mathrm{lg}} \cap \mathcal{U}_{i^*+1}| \leq b,$$
then set $i' = i^*$. Since the uncovered sets are nested, we have
$$\mathcal{U}_{i^*+1} \cap \mathcal{I} \subseteq \mathcal{I}_{\mathrm{sm}} \cup (\mathcal{I}_{\mathrm{lg}} \cap \mathcal{U}_{i^*+1}),$$
and hence
$$|\mathcal{U}_{i^*+1} \cap \mathcal{I}| \leq |\mathcal{I}_{\mathrm{sm}}| + |\mathcal{I}_{\mathrm{lg}} \cap \mathcal{U}_{i^*+1}| \leq \frac{\epsilon z}{2} + \frac{\epsilon z}{2} = \epsilon z.$$

Moreover, all points covered before layer $i^*$ are within distance at most $4L^*$ from $\mathcal{V}$ by Corollary B.3, and all points in $X_{i^*}$ are within distance at most $2L^*$ from $S_{i^*} \subseteq \mathcal{V}$. Thus
$$\delta(P \setminus \mathcal{U}_{i^*+1}, \mathcal{V}) \leq 4L^*,$$
and the claim follows in this case.

It remains to consider the case
$$|\mathcal{I}_{\mathrm{lg}} \cap \mathcal{U}_{i^*+1}| > b.$$

Since the while-loop of Algorithm 3 eventually removes all remaining points, there exists an index $j \in [i^* + 1, h]$ such that $|\mathcal{I}_{\mathrm{lg}} \cap \mathcal{U}_{j+1}| \leq b$. Let $i'$ be the smallest such index. Then, for every $j \in [i^* + 1, i']$, we have

$$|\mathcal{I}_{\mathrm{lg}} \cap \mathcal{U}_j| > b.$$

All points in $\mathcal{I}_{\mathrm{lg}}$ are within distance $2L^*$ from $S_{i^*}$. Therefore, before the batch $\mathcal{N}_j$ is selected at any residual layer $j \in [i^* + 1, i']$, the current uncovered set $\mathcal{U}_j$ contains more than $b$ points whose distance to $S_{i^*}$ is at most $2L^*$. Since $\mathcal{N}_j$ is chosen as the $b$ nearest points in $\mathcal{U}_j$ to $S_{i^*}$, every point in $\mathcal{N}_j$ also has distance at most $2L^*$ from $S_{i^*}$. Hence,

$$\delta \left( \bigcup_{j=i^*+1}^{i'} \mathcal{N}_j, S_{i^*} \right) \leq 2L^*.$$

By the choice of $i'$, we have

$$|\mathcal{I}_{\mathrm{lg}} \cap \mathcal{U}_{i'+1}| \leq b.$$

Again using the nesting of the uncovered sets,

$$\mathcal{U}_{i'+1} \cap \mathcal{I} \subseteq \mathcal{I}_{\mathrm{sm}} \cup (\mathcal{I}_{\mathrm{lg}} \cap \mathcal{U}_{i'+1}),$$

and therefore

$$|\mathcal{U}_{i'+1} \cap \mathcal{I}| \leq |\mathcal{I}_{\mathrm{sm}}| + |\mathcal{I}_{\mathrm{lg}} \cap \mathcal{U}_{i'+1}| \leq \frac{\epsilon z}{2} + \frac{\epsilon z}{2} = \epsilon z.$$

Finally, decompose the points covered by layer $i'$ as

$$P \setminus \mathcal{U}_{i'+1} = (P \setminus \mathcal{U}_{i^*}) \cup X_{i^*} \cup \bigcup_{j=i^*+1}^{i'} \mathcal{N}_j.$$

The first set is covered by type-1 representatives within distance at most $4L^*$ by Corollary B.3. The second set has distance at most $2L^*$ from $S_{i^*} \subseteq \mathcal{V}$, as shown above. The third set also has distance at most $2L^*$ from $S_{i^*} \subseteq \mathcal{V}$. Consequently,

$$\delta(P \setminus \mathcal{U}_{i'+1}, \mathcal{V}) \leq 4L^*.$$

This proves the corollary. $\qquad\qquad\qquad\qquad\qquad\qquad\qquad\qquad\qquad\qquad\qquad\qquad\qquad\qquad\qquad\quad$ $\square$

### B.2.3. THE CENTER-SELECTION STAGE

According to Corollary B.6, there exists at least one layer such that almost all inliers are covered by the union of the representatives, where the weights updated in such layer can also reflect the number of covered points. To obtain the final clustering solutions, an intuitive idea is to execute an existing $(k, z)$-center routine to decrease the number of representatives from $\tilde{O}(k/\epsilon)$ to exactly $k$ to serve as the final clustering centers, where we call this procedure the Center-Selection stage.

The Center-Selection stage operates on the weighted representatives extracted from the layered structure to construct weighted summaries (detailed in Algorithm 4). A summary consists of weighted representatives with size $\tilde{O}(k/\epsilon)$, which are constructed by combining the information from different layers. To simulate the data integration process and find the key layer index $i'$ as stated in Corollary B.6, the algorithm produces a sequence of weighted summaries and then searches for a summary under which a high-quality solution can be obtained with only $(1 + \epsilon)z$ outliers discarded.

However, there still exists a key challenge here since existing $(k, z)$-center algorithms typically rely on a radius-guessing strategy to perform the clustering process. Although effective, it could reintroduce an $O(\log \Delta)$-factor dependence on the running time. Additionally, our target in this stage is to construct solutions with bi-criteria approximations, where more than $z$ outliers can be discarded as outlier. Hence, simply enumerating pairwise distances among representatives, as in earlier greedy coverage methods (e.g., Charikar et al., 2001), may not yield good approximations in our setting.

To overcome this challenge, we propose a new center selection method, which is presented in Algorithm 4. The proposed algorithm begins by collecting all the layered structures maintained (i.e., the tuples $(\mathcal{U}_i, S_i, X_i, \rho_i, w_i)$ with indices $i \in [1, h]$). The representatives from layers with indices $i \in [i^*]$ form the point set of each summary (the set $\mathcal{V}$ in step 2 of Algorithm 4).

A candidate radii set $\mathcal{L}'$ is then constructed by collecting all the pairwise distances between the representatives in $\mathcal{V}$ and performing a discretization process on them (steps 3-7 of Algorithm 4). For each layer index $i \in [i^*, h]$, the algorithm constructs a new instance $\mathcal{I}_i'$ by updating the weights of the representatives using the weights obtained during the residual-peeling stage (steps 9-10 of Algorithm 4). At the same time, the proposed algorithm adjusts the outlier budget of the new instance (i.e., set $z' = (1+\epsilon)z - |\mathcal{U}_{i+1}|$), where points in the set $\mathcal{U}_{i+1}$ are regarded as discarded outliers in the $i$-th layer. Instead of using a $\Delta$-dependent radius searching strategy, our method restricts the binary searching for clustering radius to the constructed candidate radii set $\mathcal{L}'$ (steps 15-32 of Algorithm 4). Before executing an existing $(k,z)$-center routine to obtain the final solution (such as the 3-approximation algorithm as presented in steps 18-21 of Algorithm 4), the algorithm applies a pre-processing step, called Merging (Algorithm 5), to enforce good approximations if the representatives behave a separation property (step 11 of Algorithm 4, see Algorithm 5 for details).

Next, we give the approximation analysis for the Center-Selection Algorithm. Fixing a constructed summary $\mathcal{I}_i'$ (step 10 of Algorithm 4), there are two cases that may happen: (1) for any pair of points $p, q \in \mathcal{I}_i'$, either $\delta(p, q) \geq L^* \cdot |\mathcal{I}_i'|^3$ or $\delta(p, q) < L^*/|\mathcal{I}_i'|^3$ holds; (2) $\exists$ a pair of points $p, q \in \mathcal{I}_i'$, s.t. $L^*/|\mathcal{I}_i'|^3 \leq \delta(p, q) \leq L^* \cdot |\mathcal{I}_i'|^3$.

---

**Algorithm 4** Center-Selection$(\mathcal{D}, i^*, h, k, z, \epsilon)$

**Input:** Layered structures $\mathcal{D}$, a separating index $i^*$, the maximum layer index $h$, parameters $k$, $z$, $\epsilon$.
**Output:** A final center set $\mathcal{C}_f$.

1:  Initialize $r_f = +\infty$, $\mathcal{C}_f = \emptyset$, $\mathcal{L}' = \emptyset$.
2:  Fetch the layers $\{(\mathcal{U}_i, S_i, X_i, \rho_i, w_i)\}$ from $\mathcal{D}$ for each index $i \in [1, h]$, and set $\mathcal{V} = \bigcup_{i \in [1, i^*]} S_i$.
3:  Let $\mathcal{L}$ be the set containing all the pairwise distances between data points in $\mathcal{V}$.
4:  **for** $l \in \mathcal{L}$ **do**
5:      $\mathcal{L}' = \mathcal{L}' \cup \{(1+\epsilon)^{j'} : l/|\mathcal{V}|^3 \leq (1+\epsilon)^{j'} \leq l \cdot |\mathcal{V}|^3\} \cup \{l/|\mathcal{V}|^3\} \cup \{l \cdot |\mathcal{V}|^3\}$.
6:  **end for**
7:  Sort the candidate clustering radii in $\mathcal{L}'$ with non-decreasing order.
8:  **for** $i = i^*$ to $h$ **do**
9:      Update the weight for each point $p \in \mathcal{V}$ as $w_i(p)$.
10:     $z' \leftarrow (1+\epsilon)z - |\mathcal{U}_{i+1}|$, $\mathcal{I}_i' \leftarrow \mathcal{V}$.
11:     Call the Merging$(\mathcal{I}_i', \mathcal{L}, k, |\mathcal{U}_{i+1}|, z, \epsilon)$ algorithm to obtain a solution $(\mathcal{C}_i, r_i)$.
12:     **if** $r_i \leq r_f$ **then**
13:         $r_f = r_i$, $\mathcal{C}_f = \mathcal{C}_i$.
14:     **end if**
15:     Let $L_i$ denote the $i$-th clustering radius in $\mathcal{L}'$, and initialize $u_{id} = |\mathcal{L}'|$, $l_{id} = 1$.
16:     **while** $l_{id} \leq u_{id}$ **do**
17:         $m = \lfloor (l_{id} + u_{id})/2 \rfloor$, $\mathcal{C}' = \emptyset$, $\mathcal{U} = \mathcal{I}_i'$.
18:         **for** $j = 1$ to $k$ **do**
19:             $c_j = \arg\max_{p \in \mathcal{U}} \sum_{q \in B(p, \mathcal{U}, 10L_m)} w(q)$.
20:             $\mathcal{C}' = \mathcal{C}' \cup \{c_j\}$, $\mathcal{U} = \mathcal{U} \setminus B(c_j, \mathcal{U}, 20L_m)$.
21:         **end for**
22:         $z'' = \sum_{p \in \mathcal{U}} w(p)$.
23:         **if** $z'' \leq z'$ **then**
24:             **if** $4L_m < r_f$ **then**
25:                 $r_f = 20L_m$, $\mathcal{C}_f = \mathcal{C}'$.
26:             **end if**
27:             $u_{id} = m - 1$.
28:         **else**
29:             $l_{id} = m + 1$.
30:         **end if**
31:     **end while**
32: **end for**
33: **return** $\mathcal{C}_f$.

---

If case (2) happens, it is obvious that we can construct a set of possible clustering radii based on all the pairwise distances with a fine-grained division strategy (steps 3-7 of Algorithm 4) to well approximate the optimal clustering radius. Let $\mathcal{L}'$ be

the set of candidate radii obtained through steps 3-7 of Algorithm 4, the following lemma shows that there exists at least one radius in $\mathcal{L}'$ that can well approximate the optimal clustering radius $L^*$.

**Lemma B.7.** *If case (2) happens, there exists at least one radius $L' \in \mathcal{L}'$ such that $L^* \leq L' \leq (1+\epsilon)L^*$.*

*Proof.* Let $p, q \in \mathcal{I}'_i$ be the pair of points in instance $\mathcal{I}'_i$ satisfying $L^*/|\mathcal{I}'_i|^3 \leq \delta(p, q) \leq L^* \cdot |\mathcal{I}'_i|^3$. We can easily get that $L^*$ lies in the range $[\delta(p, q)/|\mathcal{I}'_i|^3, |\mathcal{I}'_i|^3 \cdot \delta(p, q)]$. By further dividing the range into logarithmic smaller blocks of $\{(1+\epsilon)^j : \delta(p, q)/|\mathcal{I}'_i|^3 \leq (1+\epsilon)^j \leq |\mathcal{I}'_i|^3 \cdot \delta(p, q))\}$, we can get that there exists at least one radius in $\mathcal{L}'$ that can well approximate the value for $L^*$ with only $(1+\epsilon)$-approximation loss. Since the Center-Selection algorithm iterates over all the pairwise distances, the pair $p, q$ can be identified, which proves Lemma B.7. $\square$

Lemma B.7 implies that there is an approximate optimal radius in the candidate radii set $\mathcal{L}'$. To find such radius, a $k$-center with outliers solver can be invoked to test the feasibility of the radius together with the clustering solutions obtained based on this radius. Before formal analysis, we should first give the definitions of feasible weighted summaries. Recall that each weighted summary $\mathcal{I}'_i$ is constructed by combining with the tuple $(\mathcal{U}_i, S_i, X_i, \rho_i, w_i)$ in the $i$-th layer. In general, a weighted summary is called feasible if the covering radius induced by the representatives in the summary can be bounded by a constant factor of the optimal clustering and the number of uncovered inliers can also be bounded by $\epsilon z$.

**Definition B.8.** A weighted summary $\mathcal{I}'_i$ constructed based on the tuple of the $i$-th layered structure $(\mathcal{U}_i, S_i, X_i, \rho_i, w_i)$ is called feasible if it satisfies that: (1) $\delta(P \setminus (\mathcal{U}_i \setminus X_i), \mathcal{V}) \leq \beta L^*$ for some constant $\beta$; (2) $|(\mathcal{U}_i \setminus X_i) \cap \mathcal{I}| \leq \epsilon z$.

Indeed, steps 18–21 of Algorithm 4 implement a standard clustering routine, which is obtained by adapting a classic deterministic 3-approximation algorithm (Charikar et al., 2001) to the weighted setting. It has been proved in previous literature (Li & Guo, 2018) that, given any feasible clustering instance $\mathcal{I}'_i$, the greedy clustering routine can roughly return a $(5\beta + 4)$-approximate solution with $(1+\epsilon)z$ outliers discarded.

**Lemma B.9.** (Li & Guo, 2018) *Given a feasible weighted summary $\mathcal{I}'$ with $\beta$-approximation on clustering quality (where $\beta = 4$), then the clustering routine in steps 18-21 of Algorithm 4 can return a set $\mathcal{C}'$ of clustering centers with $((5\beta + 4)(1 + \epsilon))$-approximation, while the overall uncovered points can be bounded by $(1 + \epsilon)z$.*

Lemma B.9 indicates that whenever case (2) happens, our algorithm can always obtain a feasible solution with $(1+\epsilon)z$ outliers discarded. Combining with Corollary B.6, we have that the approximation ratio induced by the feasible weighted summary is 4, where a $24(1 + \epsilon)$-approximate solution with $(1 + \epsilon)z$ outliers discarded can be obtained. Then, we consider a more complicated case (i.e., case (1)). In this case, it is also obvious that the pairwise distances can no longer be leveraged to construct candidate clustering radii that can well approximate the optimal one. However, a key observation is that the representatives in the summary satisfy a separation property: any two representatives are either close enough to be merged, or they are well separated from each other. Based on such observation, we propose a pre-processing algorithm (i.e., Algorithm 5) that can be used to obtain feasible solutions with good approximation guarantees.

More specific, given a weighted summary $\mathcal{I}'$, a set $\mathcal{L}$ (as the input for Algorithm 5) consisting of all the pairwise distances between data points in $\mathcal{I}'$ (constructed in step 3 of Algorithm 4) is used to serve as candidate radii. To handle case (1) condition, the algorithm performs a binary search (steps 4-24 of Algorithm 5) over the sorted candidate radii in $\mathcal{L}$. For a fixed candidate radius $L_m \in \mathcal{L}$, the algorithm then conducts a representative merging process (steps 6-13 of Algorithm 5) that integrates data points close to each other. Starting with the full representative set as the uncovered points (step 6 of Algorithm 5), the algorithm traverses each uncovered representative. Whenever an uncovered representative $p$ is visited (step 7 of Algorithm 5), the algorithm treats $p$ as a temporary center and merges all the nearby uncovered points to it (step 10 of Algorithm 5), and finally assigns $p$ a weight equal to the total weight of the covered points (step 12 of Algorithm 5). After that, all such nearby covered points are removed from the set of uncovered representatives. This process produces a set of new representatives (i.e., the set $\mathcal{I}''$), with each assigned a weight denoting the covered weight summations.

After this procedure, the algorithm selects the top-$k$ representatives with the largest weights as the final clustering centers (step 14 of Algorithm 5). The remaining uncovered weight summations are then interpreted as the number of discarded outliers (step 15 of Algorithm 5). If the uncovered weight summations does not exceed the adjusted outlier budget, the algorithm records the corresponding center set as a valid solution (steps 16-19 of Algorithm 5).

According to Corollary B.6, there always exists a feasible weighted summary. Then, based on this feasible weighted summary $\mathcal{I}'_i$, we show that the resulting weighted instance $\mathcal{I}''$ constructed in steps 6-13 of Algorithm 5 guarantees that selecting the top-$k$ representatives with the largest weights induces a feasible solution with bounded approximation guarantees on both

---

**Algorithm 5** Merging$(\mathcal{I}', \mathcal{L}, k, z', z, \epsilon)$

---

**Input:** A weighted summary $\mathcal{I}'$ of data points where each $p \in \mathcal{I}'$ is associated with a weight $w_i'(p)$, a set $\mathcal{L}$ of candidate clustering radii, parameters $k, z', z$ and $\epsilon$.

**Output:** A set $\mathcal{C}_0$ of clustering centers, a clustering radius $r_0$.

1: $r_0 = +\infty, \mathcal{C}_0 = \emptyset, z_0 = +\infty$.
2: Set $\mathcal{L} = \mathcal{L} \cup \{0\}$, and sort the radius in $\mathcal{L}$ with non-decreasing order.
3: Let $L_m$ denote the $m$-th clustering radius in $\mathcal{L}$, and initialize $u_{id} = |\mathcal{L}|, l_{id} = 1$.
4: **while** $l_{id} \leq u_{id}$ **do**
5:     $m = \lfloor (l_{id} + u_{id})/2 \rfloor$.
6:     Initialize $\mathcal{I}'' = \emptyset, \mathcal{U} = \mathcal{I}'$.
7:     **for** $p \in \mathcal{U}$ **do**
8:         $j \leftarrow 1, \mathcal{U} \leftarrow \mathcal{U} \backslash \{p\}, \mathcal{M}(p) = \{p\}$.
9:         **while** $B(p, \mathcal{U}, L_m \cdot j) \neq \emptyset$ **do**
10:             $\mathcal{M}(p) \leftarrow \mathcal{M}(p) \cup B(p, \mathcal{U}, L_m \cdot j), \mathcal{U} \leftarrow \mathcal{U} \backslash B(p, \mathcal{U}, L_m \cdot j), j \leftarrow j + 1$.
11:         **end while**
12:         $\mathcal{I}'' \leftarrow \mathcal{I}'' \cup \{p\}$, assign a weight $w(p) = \sum_{q \in \mathcal{M}(p)} w_i'(q)$ to $p$.
13:     **end for**
14:     Let $\mathcal{C}'$ be the set of the data points in $\mathcal{I}''$ with top-$k$ weights.
15:     $z'' = z' + \sum_{p \in \mathcal{I}'' \backslash \mathcal{C}'} w(p)$.
16:     **if** $z'' \leq (1 + \epsilon)z$ **then**
17:         **if** $|\mathcal{I}'| \cdot L_m < r_0$ **then**
18:             $r_0 = |\mathcal{I}'| \cdot L_m, \mathcal{C}_0 = \mathcal{C}'$.
19:         **end if**
20:         $u_{id} = m - 1$.
21:     **else**
22:         $l_{id} = m + 1$.
23:     **end if**
24: **end while**
25: **return** $\mathcal{C}_0, r_0$.

---

the clustering quality and the number of discarded outliers. Let $\mathcal{I}_i'$ be any feasible weighted instance derived from the $i$-th layered structure $(\mathcal{U}_i, S_i, X_i, \rho_i, w_i)$. Recall that $\mathcal{L}$ is the set of pairwise distances of points in $\mathcal{I}_i'$. We define $l'$ as the radius threshold such that $l' = \max\{l \in \mathcal{L} : l < L^*/|\mathcal{I}_i'|^3\}$. For any given searching radius $L_m \geq l'$ (step 5 of Algorithm 5, the radius $L_m$), a feasible clustering solution that discards at most $(1 + \epsilon)z$ outliers can be constructed by our Algorithm 5.

**Lemma B.10.** *Let $\mathcal{I}_i' = (\mathcal{V}, w_i)$ be a 4-feasible weighted summary, and let $M = |\mathcal{V}|$. Suppose that $\mathcal{V}$ satisfies the gap condition that for every two distinct representatives $u, v \in \mathcal{V}$, $\delta(u, v) < \frac{L^*}{M^3}$ or $\delta(u, v) \geq M^3 L^*$. Let $\ell^* = \max\left(\{0\} \cup \{\delta(u, v) : u, v \in \mathcal{V}, u \neq v, \delta(u, v) < L^*/M^3\}\right)$. Then Algorithm 5, when run on the radius $\ell^*$, constructs a center set $\mathcal{C}'$ whose $(1 + \epsilon)z$-outlier radius on $P$ is at most $5L^*$.*

*Proof.* Let $\sigma_i : P \setminus \mathcal{U}_{i+1} \to \mathcal{V}$ be the assignment map of the feasible summary. Since the summary is 4-feasible, every represented point $p \in P \setminus \mathcal{U}_{i+1}$ satisfies

$$\delta(p, \sigma_i(p)) \leq 4L^*,$$

and the number of uncovered true inliers satisfies

$$|\mathcal{U}_{i+1} \cap \mathcal{I}| \leq \epsilon z.$$

Consider an arbitrary optimal cluster $P_h^*$ and two represented inliers $p, q \in P_h^* \setminus \mathcal{U}_{i+1}$. Since $p$ and $q$ belong to the same optimal cluster,

$$\delta(p, q) \leq 2L^*.$$

Therefore,

$$\delta(\sigma_i(p), \sigma_i(q)) \leq \delta(\sigma_i(p), p) + \delta(p, q) + \delta(q, \sigma_i(q)) \leq 10L^*.$$

Assume $M > 3$. Otherwise the instance has constant summary size and can be handled by exhaustive enumeration. Under the gap condition, the distance $\delta(\sigma_i(p), \sigma_i(q))$ cannot be at least $M^3 L^*$, because $M^3 L^* > 10 L^*$. Hence,

$$\delta(\sigma_i(p), \sigma_i(q)) < \frac{L^*}{M^3} \leq \ell^*.$$

Thus, for each optimal cluster $P_h^*$, all representatives assigned to represented inliers of $P_h^*$ are pairwise within distance $\ell^*$.

Now consider the merging procedure with radius $\ell^*$. For each optimal cluster $P_h^*$ whose represented inlier set is nonempty, let $r_h$ be the first representative assigned to this cluster that is processed by the merging procedure. Since every other representative assigned to represented inliers of $P_h^*$ is within distance $\ell^*$ of $r_h$, all such representatives are merged into the part represented by $r_h$. Hence the set of representatives

$$R = \{r_h : P_h^* \setminus \mathcal{U}_{i+1} \neq \emptyset\}$$

has size at most $k$ and covers total weight at least

$$|\mathcal{I} \setminus \mathcal{U}_{i+1}| \geq (n - z) - \epsilon z = n - (1 + \epsilon)z.$$

Since the total weight in the summary is $n - |\mathcal{U}_{i+1}|$, the $k$ heaviest representatives in the merged instance cover weight at least $n - (1 + \epsilon)z$. Therefore the uncovered summary weight is at most

$$(n - |\mathcal{U}_{i+1}|) - (n - (1 + \epsilon)z) = (1 + \epsilon)z - |\mathcal{U}_{i+1}|.$$

This is exactly the residual outlier budget $z'$ used in Algorithm 5. Hence the candidate center set is feasible with respect to the allowed number of outliers.

Finally, every representative merged into a chosen merged center is at distance at most $M\ell^*$ from that center, because the merging radius can increase for at most $M$ rounds. Thus every original point represented by a chosen merged center is at distance at most

$$4L^* + M\ell^* \leq 4L^* + M \cdot \frac{L^*}{M^3} \leq 5L^*$$

from the selected center. The points not represented by chosen merged centers, together with the points in $\mathcal{U}_{i+1}$, have total cardinality at most $(1 + \epsilon)z$. Therefore the selected centers have $(1 + \epsilon)z$-outlier radius at most $5L^*$. $\square$

According to Lemma B.10, by combining with binary search strategies on the sorted pairwise distances, if the given weighted instance $\mathcal{I}_i'$ satisfies the gap condition (case (2) condition), Algorithm 5 can return a feasible solution with bounded radius and $(1 + \epsilon)z$ outliers discarded. Let $\mathcal{C}'$ be the set of centers constructed in step 14 of Algorithm 5 using $l' = \max\{l \in \mathcal{L} : l < L^*/|\mathcal{I}_i'|^3\}$ as the searching radius. Denote $\mathcal{Y}$ as the set of data points whose representatives are merged to the centers in $\mathcal{C}'$. Then, we have $\delta(p, \mathcal{C}') \leq 4L^* + l' \cdot |\mathcal{I}_i'| \leq 5L^*$ using triangle inequality. According to Lemma B.10, the overall uncovered weight units for $\mathcal{C}'$ can be bounded by $(1 + \epsilon)z$, which implies that $\mathcal{C}'$ yields a 5-approximation with $(1+\epsilon)z$ outliers discarded. Since the binary search process returns the feasible solution with minimum searching radius, Algorithm 5 can return a 5-approximation if the given weighted summary $\mathcal{I}_i'$ satisfies case (2) conditions.

Together with Lemma B.9, Theorem 1 can be proved.

**Theorem 3.1** *With constant probability, Layered-Sampling can return a $24(1 + \epsilon)$-approximate solution in time $\tilde{O}(nk/\epsilon^2 + k^2/\epsilon^4)$ with $(1 + \epsilon)z$ outliers discarded.*

*Proof.* The approximation guarantees on clustering quality follows from Corollary B.6, Lemma B.9 and Lemma B.10, where a $24(1 + \epsilon)$-approximation can be obtained. Then, we analyze the running time.

As shown in the literature (Chen et al., 2018), the Greedy-Coverage stage takes time $O(nk \log n)$. In our analysis, we assume $k \geq \log n$, and hence $\max\{k, \log n\} = k$. Therefore, the sampling size used in the Greedy-Coverage stage is $O(k)$, and the total cost of this stage can be bounded by $O(nk \log n)$.

For the Residual-Peeling stage, the sampling step takes lower-order time compared with the subsequent assignment and selection steps. The number of sampled representatives in this stage is $O(k \log k/\epsilon)$. Finding the farthest $(1 + \epsilon)z$ uncovered points to the representatives and assigning data points to their closest representatives can be executed in time $O(nk \log k/\epsilon)$

using the linear selection method (Blum et al., 1973). Since the number of uncovered points is smaller than $8z$ before entering the Residual-Peeling stage and $\epsilon z/2$ points are incorporated in each iteration, there are $O(1/\epsilon)$ iterations for the while loop of steps 5–10 of Algorithm 3. In each iteration, finding the nearest points and updating the data assignments can also be executed in time $O(nk \log k/\epsilon)$ using the linear selection method (Blum et al., 1973). Hence, the overall running time for the Residual-Peeling stage can be bounded by $O(nk \log k/\epsilon^2)$.

For the Center-Selection stage, let $M = |\mathcal{V}|$ denote the number of representatives. Since $\max\{k, \log n\} = k$, we have $M = O(k \log n + k \log k/\epsilon) = O(k(\log n + \log k/\epsilon))$. Calculating the pairwise distances for constructing the candidate radii set $\mathcal{L}$ takes time $O(M^2)$, which is $O(k^2(\log n + \log k/\epsilon)^2)$. Then, for each radius in $\mathcal{L}$, the algorithm generates $O(\log_{1+\epsilon}(|\mathcal{V}|^6)) = O(\log M/\epsilon)$ candidate radii. Since $|\mathcal{L}| = O(M^2)$, we have $|\mathcal{L}'| = O(M^2 \log M/\epsilon)$. Therefore, constructing and sorting $\mathcal{L}'$ takes time $O((M^2 \log M/\epsilon) \log(M/\epsilon))$.

There are $O(1/\epsilon)$ iterations in the for loop of step 8 in Algorithm 4. In each iteration, Algorithm 4 performs a binary search over the sorted candidate radii set $\mathcal{L}'$, which takes $O(\log(M/\epsilon))$ search steps. For each guessed radius, the accelerated greedy clustering routine can be implemented in time $O(M^2)$, following the same idea as the greedy coverage routine used in (Chen et al., 2018). Hence, the clustering routine in one iteration takes time $O(M^2 \log(M/\epsilon))$, and over all $O(1/\epsilon)$ iterations it takes time $O((M^2/\epsilon) \log(M/\epsilon))$.

As for the merging process, the sorting step and the subsequent merging operations can be bounded by $O(M^2 \log M)$ per iteration. Together with the binary searching strategy, the total time complexity of the merging process over all $O(1/\epsilon)$ iterations is bounded by $O((M^2 \log M/\epsilon) \log(M/\epsilon))$. This term also dominates the accelerated greedy clustering cost. Therefore, the overall running time for the Center-Selection stage can be bounded by $O((M^2 \log M/\epsilon) \log(M/\epsilon))$.

Substituting $M = O(k \log n + k \log k/\epsilon)$, the running time of the Center-Selection stage is bounded by

$$O\left( \frac{\left( k \log n + \frac{k \log k}{\epsilon} \right)^2}{\epsilon} \log\left( k \log n + \frac{k \log k}{\epsilon} \right) \log\left( \frac{k \log n + \frac{k \log k}{\epsilon}}{\epsilon} \right) \right).$$

Equivalently, this can be written as

$$O\left( \frac{k^2}{\epsilon} \left( \log n + \frac{\log k}{\epsilon} \right)^2 \log\left( k \log n + \frac{k \log k}{\epsilon} \right) \log\left( \frac{k \log n + \frac{k \log k}{\epsilon}}{\epsilon} \right) \right).$$

Putting the Greedy-Coverage, Residual-Peeling, and Center-Selection stages together, the total running time of the Layered-Sampling algorithm can be bounded by

$$O\left( nk \log n + \frac{nk \log k}{\epsilon^2} + \frac{k^2}{\epsilon} \left( \log n + \frac{\log k}{\epsilon} \right)^2 \log\left( k \log n + \frac{k \log k}{\epsilon} \right) \log\left( \frac{k \log n + \frac{k \log k}{\epsilon}}{\epsilon} \right) \right).$$

If we additionally use the standard assumption $k \le n$, then $\log k \le \log n$, and the bound can be simplified to

$$O\left( nk \log n + \frac{nk \log k}{\epsilon^2} + \frac{k^2 \log^2 n}{\epsilon^3} \log\left( \frac{k \log n}{\epsilon} \right) \log\left( \frac{k \log n}{\epsilon^2} \right) \right).$$

Using $\tilde{O}$ notations, we can get that the overall time complexity can be bounded by $\tilde{O}(nk/\epsilon^2 + k^2/\epsilon^4)$. □

## C. The Dynamic $(k, z)$-center Algorithm

In this section, we provide a detailed analysis for the proposed dynamic $(k, z)$-center algorithm (i.e., Algorithm 2 in Section 3.2). The main idea behind is to maintain similar layered structures as in static settings during the updates. To achieve fast update time, a key strategy is that the dynamic structure is rebuilt only after a certain accumulation of operations. In the following, for better consistency, we also list the properties for the dynamic structure $\mathcal{D}'$ as presented in the main context.

- The data structure $\mathcal{D}'$ maintains a collection of $h$ layers. These layers are further divided into three types by a separating index $i^* < h$. Layers with indices $j < i^*$ form the type-1 layer, the layer with index $j = i^*$ forms the type-2 layer (also denoted as the transition layer), and layers with indices $j > i^*$ form the type-3 layer.

- Each layer is represented by a tuple $(\mathcal{U}_i, S_i, X_i, \rho_i, w_i)$ (for some index $i \leq h$), where $\mathcal{U}_i$ denotes the set of currently uncovered data points, $S_i$ is the set of representatives in the $i$-th layer, $X_i \subseteq \mathcal{U}_i$ denotes the set of points covered in the $i$-th layer by balls of radius $\rho_i$ centered at representatives in $S_i$, and $w_i : S_i \to \mathbb{R}$ is a mapping function that assigns a weight $w_i(p)$ to each $p \in S_i$.

- It is guaranteed that for each type of layer, the covered point set takes a certain fraction of the uncovered point set, i.e., $|X_i| = \alpha_i |\mathcal{U}_i|$ holds for some $0 < \alpha_i < 1$.

- Each type-1 and type-2 layer maintains a counter $n_i$, which denotes the number of updates to $\mathcal{U}_i$ since the last time $\mathcal{U}_i$ is rebuilt. For each layer with index $i < i^*$, let $\mathcal{U}_i^{\text{OLD}}$ denote the state of the uncovered set in the $i$-th layer since the last time it was reconstructed. The counter is used to trigger the updates of the layered structure. There is a parameter $0 < \tau < 1$ such that $n_i \leq \tau |\mathcal{U}_i^{\text{OLD}}|$ holds for each layer with index $i \in [1, i^*]$.

- The data structure $\mathcal{D}'$ also maintains a set $\mathcal{Z}$ of candidate outliers during the whole dynamic update processes.

Similar to the case in static settings, the layered structure $\mathcal{D}'$ is also constructed to separate the instance into inlier parts and candidate outlier parts. However, different from static settings, a counter is maintained for each layer to record the accumulated updates. For a specific layer maintained, once the accumulated updates reach a fixed fraction of the uncovered data size, the layered structure is rebuilt from this layer. This update-delay strategy aims to avoid triggering updates too frequently and is crucial for achieving sub-linear update time in the fully-dynamic setting.

## C.1. High-Level Description of the Update Strategies

Before presenting the formal analysis, we first give a high-level description of the main update procedure.

**Handling the Insertion:** After receiving an operation for inserting a point $x$, the algorithm first adds the point $x$ to $\mathcal{U}_i$ for each type-1 layer with indices $i \in [i^* - 1]$, where $i^*$ is the separating index that distinguishes different types of layers. Meanwhile, the counters (i.e., $n_i$) for these layers are also updated to record how many changes have been made since the last rebuild. Then, we need to consider the possibility that the inserted point could be selected as a center in the $i^*$-th layer. To maintain the correct sampling distribution, a random replacement process is performed on $x$ to simulate that $x$ is randomly chosen from $\mathcal{U}_{i^*} \cup \{x\}$ using a Bernoulli distribution. If $x$ is selected as one of the centers in $S_{i^*}$, then we reconstruct the $i^*$-th layers to maintain the insertion invariant. Otherwise, the algorithm treats $x$ as a candidate outlier and inserts it into the uncovered sets $\mathcal{U}_i$ for all $i \in [i^*, h]$ as well as the outlier candidate set $\mathcal{Z}$. Once the counters indicate that a certain fraction of updates (i.e., $n_i \geq \tau \cdot |\mathcal{U}_i^{\text{OLD}}|$ for some parameter $0 < \tau < 1$) have been made for a specific layer, then the algorithm reconstructs the layered structures from such layer and reset the counter.

**Handling the Deletion:** After receiving an operation for deleting a point $x$, it will be removed from the set $\mathcal{U}_i$ of uncovered data points for each layer with index $i \in [h]$ if $x$ belongs to $\mathcal{U}_i$. Then, there are four cases that may happen: (1) $x$ is a representative in $S_i$ for some layer with index $i < i^*$; (2) $x$ is a representative in the transition layer; (3) $x$ is a covered point in $X_i$ for some layer index $i < h$; (4) $x$ belongs to the set $\mathcal{Z}$ of candidate outliers. If case (1) happens, the point $x$ is removed from $S_i$, and an alternative $x' \in S_i$ with $x' \neq x$ and $\sigma(x) = \sigma(x')$ is selected to replace $x$ as a new representative. If case (2) happens, then the layers are rebuilt from the $i^*$-th layer (i.e., the transition layer) since the current representative set $S_{i^*}$ may no longer follow a uniform sampling distribution. Finally, if case (3) or case (4) happens, $x$ is removed directly from $X_i$ or $\mathcal{Z}$. We will show that such removal has little impact on the approximation guarantees since $x$ is not a representative. Meanwhile, for the layered structures maintained, the algorithm updates the counter $n_i$ for $i \in [i^*]$ to record how many changes have been made since the last rebuild. Once the counters indicate that a certain fraction of updates (i.e., $n_i \geq \tau \cdot |\mathcal{U}_i^{\text{OLD}}|$ for some parameter $0 < \tau < 1$) have been made for a specific layer, then the algorithm reconstructs the layered structures from such layer and reset the counter.

In the following, we discuss the detailed approximation analysis. Upon receiving an initial dataset $P$, the proposed algorithm calls the static algorithm (Algorithm 1) to construct a sequence of initial layered structures $(\mathcal{U}_i, S_i, X_i, \rho_i, w_i)$ for $i \in [h]$, where $h$ is the total number of layers obtained (step 1 of Algorithm 2). Meanwhile, for each layer index $i \in [h]$, the counter is initialized as $n_i = 0$, and the set $\mathcal{Z}$ is initialized to be an empty set (step 2 of Algorithm 2).

## C.2. Algorithmic Descriptions for Insertion and Deletion Operations

The formal description for insertion algorithm is described in Algorithm 6. Upon inserting a new point $x$, the algorithm simply regards it as an uncovered data point in layers with indices $i < i^*$. The algorithm begins by adding $x$ to the uncovered

sets $\mathcal{U}_i$ for all layers with indices $i < i^*$ (step 2 of Algorithm 6), and increases the corresponding counters $n_i$ (step 3 of Algorithm 6). Next, the algorithm considers the possibility that $x$ is selected as a representative in the $i^*$-th layer (i.e., the transition layer). To maintain sampling distributions, the algorithm performs a Bernoulli trial with $|S_{i^*}|$ independent sampling steps, where the sampling probability is set as $\frac{1}{|\mathcal{U}_{i^*}|+1}$ (steps 5-11 of Algorithm 6), where the intuitive idea behind is to simulate that $\Theta(k \log k/\epsilon)$ representatives can be uniformly and independently sampled from the candidate set $\mathcal{U}_{i^*} \cup \{x\}$. If one of the Bernoulli trial succeeds (reflected by counter $c_x$), it indicates that the representative set $S_{i^*}$ has changed and the algorithm invokes the Residual-Peeling procedure to reconstruct the layers built on top of the $i^*$-th layer. Otherwise, the inserted data point $x$ is added to each uncovered set $\mathcal{U}_i$ for $i \in [i^*, h]$ and the set $\mathcal{Z}$ of candidate outliers (steps 15-16 and step 19 of Algorithm 6). Finally, the algorithm increases the corresponding counter $n_i^*$ (step 18 of Algorithm 6) and calls the Rebuild routine to update the layered structure according to the counters (step 21 of Algorithm 6).

---

**Algorithm 6** Insert$(x, \tau)$

---

1: **for** $i = 1$ to $i^* - 1$ **do**
2:      Add $x$ to $\mathcal{U}_i$.
3:      $n_i \leftarrow n_i + 1$.
4: **end for**
5: $c_x = 0$.
6: **for** $s \in S_{i^*}$ **do**
7:      Sample Bernoulli $I$, s.t. $\mathbf{Pr}[I = 1] = \frac{1}{|\mathcal{U}_{i^*}|+1}$.
8:      **if** $I = 1$ **then**
9:          Replace $s$ with $x$, $c_x \leftarrow c_x + 1$.
10:      **end if**
11: **end for**
12: **if** $c_x > 0$ **then**
13:      Set $\mathcal{Z} = \emptyset$, $n_{i^*} = 0$, fix $S_{i^*}$ and call a residual-peeling algorithm with fixed center set version to construct the layers starting from the $i^*$-th layer.
14: **else**
15:      **for** $i = i^*$ to $h$ **do**
16:          Add $x$ to $\mathcal{U}_i$.
17:      **end for**
18:      $n_{i^*} \leftarrow n_{i^*} + 1$.
19:      $\mathcal{Z} \leftarrow \mathcal{Z} \cup \{x\}$.
20: **end if**
21: Call the Rebuild$(\tau)$ algorithm to update the layered structure.

---

**Algorithm 8** Rebuild$(\tau)$

---

1: $i \leftarrow 1$.
2: **while** $i \leq i^*$ and $n_i \leq \tau |\mathcal{U}_i^{\text{OLD}}|$ **do**
3:      $i \leftarrow i + 1$.
4: **end while**
5: **if** $i \leq i^*$ **then**
6:      Set $\mathcal{Z} = \emptyset$.
7:      ConstructFromLayer$(i)$.
8: **end if**

---

---

**Algorithm 7** Delete$(x, \tau)$

---

1: **for** $i = 1$ to $i^* - 1$ **do**
2:     **if** $x \in \mathcal{U}_i$ **then**
3:         Remove $x$ from $\mathcal{U}_i$, and set $n_i = n_i + 1$.
4:         **if** $x \in X_i$ **then**
5:             Remove $x$ from $X_i$.
6:             **if** $x \in S_i$ **then**
7:                 Remove $x$ from $S_i$.
8:                 Denote $W_i = \{y \in X_i : \sigma(y) = x\}$ as the set of covered data points in $X_i$ that are assigned to $x$, and add a random point $r \in W_i$ to $S_i$.
9:                 Set $\sigma(y) = r$ for each $y \in W_i$ and $w_i(r) = w_i(x) - 1$, and update $\rho_i$.
10:             **end if**
11:         **end if**
12:     **end if**
13: **end for**
14: **if** $x \in S_{i^*}$ **then**
15:     Set $\mathcal{Z} = \emptyset$, $n_{i^*} = 0$, and call Residual-Peeling$(i^*, \mathcal{U}_{i^*} \setminus \{x\}, k, z, \epsilon)$.
16: **else**
17:     If $x$ belongs to $\mathcal{Z}$, then remove $x$ from $\mathcal{Z}$.
18:     If $x$ belongs to $\mathcal{U}_{i^*}$, set $n_i^* = n_i^* + 1$.
19:     **for** $i = i^*$ to $h$ **do**
20:         **if** $x \in \mathcal{U}_i$ **then**
21:             Remove $x$ from $\mathcal{U}_i$.
22:             **if** $x \in X_i$ **then**
23:                 Remove $x$ from $X_i$, and set $w_j(\sigma(x)) = w_j(\sigma(x)) - 1$ for each $j \geq i$.
24:             **end if**
25:         **end if**
26:     **end for**
27: **end if**
28: Call the Rebuild$(\tau)$ algorithm to update the layered structure.

---

**Algorithm 9** ConstructFromLayer$(j)$

---

**Input:** A layer index $j$
**Output:** Updated layered structures.

1: $i \leftarrow j, \mathcal{C} = \emptyset$.
2: **while** $|\mathcal{U}_i| > 8z$ **do**
3:     Sample a set $S_i$ of $\Theta(\xi)$ points randomly and independently from $\mathcal{U}_i$, where $\xi = \max\{k, \log |\mathcal{U}_j|\}$.
4:     Let $\rho_i$ be the smallest radius s.t. $|B(S_i, \mathcal{U}_i, \rho_i)| \geq \alpha |\mathcal{U}_i|$.
5:     $X_i \leftarrow B(S_i, \mathcal{U}_i, \rho_i), \mathcal{C} \leftarrow \mathcal{C} \cup S_i$.
6:     Assign each $p \in X_i$ to its nearest center $\sigma(p) = \arg\min_{s \in S_i} \delta(p, s)$, and set $w_i(s) = \big|\{p \in X_i : \sigma(p) = s\}\big|$ for each $s \in S_i$ .
7:     $\mathcal{U}_{i+1} \leftarrow \mathcal{U}_i \setminus X_i, n_i \leftarrow 0, i \leftarrow i + 1$.
8: **end while**
9: $i^* \leftarrow i, n_{i^*} \leftarrow 0, \mathcal{Z} = \emptyset$.
10: Call Residual-Peeling$(i^*, \mathcal{U}_{i^*}, k, z, \epsilon)$ algorithm to obtain the maximum layer index $h$, and a sequence of layered structures $(\mathcal{U}_j, S_j, X_j, \rho_j, w_j)$ for $j \in [i^*, h]$.
11: Construct $\mathcal{D}' = \{(\mathcal{U}_j, S_j, X_j, \rho_j, w_j) : j \in [1, h]\}$.

---

For the deletion operation, the formal description of the algorithm is described in Algorithm 7. Upon receiving an operation for deleting a point $x$, the algorithm updates the layered structure according to the role of $x$ in the layered structures. For layers with indices $i \in [1, i^* - 1]$, if $x$ appears in the uncovered set $\mathcal{U}_i$, the algorithm removes $x$ from $\mathcal{U}_i$ and increases

the counter $n_i$ (steps 2-3 of Algorithm 7). If $x$ further belongs to the covered set $X_i$, it is removed from $X_i$ (steps 4-5 of Algorithm 7). The only non-trivial case is when $x$ is selected as a representative in $S_i$. In this case, the algorithm removes $x$ from $S_i$, and then collects the set $W_i$ of points that were assigned to $x$ since the last update (step 8 of Algorithm 7). Note that the representative $x$ will be removed, a replacement should be selected. To handle this case, a random data point $r \in W_i$ is then selected from $W_i$ (step 8 of Algorithm 7), where the assignment is updated correspondingly by redirecting every point in $W_i$ to $r$ (step 9 of Algorithm 7). This replacement rule maintains the approximate coverage radius for each layer while avoiding a full reconstruction.

After processing the layered structures with indices $i < i^*$, the algorithm checks whether $x$ is a representative in the $i^*$-th layer (i.e., the transition layer). If $x \in S_{i^*}$, it indicates that the representative set has changed and some data points may miss their representatives with unbounded coverage radius. To address this issue, the algorithm invokes the Residual-Peeling procedure to reconstruct the layers with indices larger than $i^*$ (steps 14–15 of Algorithm 7). Otherwise, similar to the cases for earlier layers with indices smaller than $i^*$, the algorithm simply removes $x$ from the uncovered set $\mathcal{U}_i$ (if present) and increases the counter $n_i^*$ (step 21 of Algorithm 7). Additionally, if $x$ also appears in the covered set $X_i$ or the candidate outlier set $\mathcal{Z}$, it is removed from these sets as well (step 17 and step 23 of Algorithm 7). Finally, the Rebuild routine is called to update the layered structure according to counters maintained (step 28 of Algorithm 7).

Finally, we present the rebuild procedure, which is detailed in Algorithm 8. The Rebuild procedure iterates over all indices $i \in [1, i^*]$ to check whether the maintained counters reach their thresholds. Once such a layer index $j$ exists, the layered structure is rebuilt from this layer using the static algorithm (Algorithm 1) with the uncovered point set $\mathcal{U}_j$ as the input (the ConstructFromLayer algorithm in Algorithm 9).

### C.3. The Approximation Analysis

To begin the analysis, we first show that the number of layers maintained by our algorithm is comparable with the static settings. The proofs are based on a key observation about the upper bounds of the counters maintained. In the following, we fix real numbers $v$ and $\epsilon'$ such that $\alpha < v < 1$, $0 < \epsilon' < \min\{(1-v)/(2v), 1\}$ and $0 < \epsilon' < 1/4$. Note that we also require the parameter $\tau$ (associated with counter threshold) to satisfy that $\tau \le \epsilon'\alpha \le 1/3$ and $\tau \le \epsilon/16$.

**Invariant 3.2.** *Given a parameter $0 < \tau < 1$, it holds that $n_i \le \tau|\mathcal{U}_i^{\mathrm{OLD}}|$ for each $i \in [1, i^*]$.*

Note that Invariant 3.2 holds trivially since after each insertion and deletion operation, the rebuild procedure is invoked to maintain this invariant. Based on this key invariant, we can argue that the number of layers maintained by the dynamic algorithm can be bounded by $\tilde{O}(1/\epsilon)$ using the following math tool.

**Claim C.1.** (Bhattacharya et al., 2023) *Given some reals $A \ge a \ge 0$, $0 \le N \le A - a$, and define a function $f : [0, 1] \to \mathbb{R}$ as $f(x) = (a + xN)/(A - (1 - x)N)$, it holds that $f(x) \le f(1)$ for all $x \in [0, 1]$.*

**Lemma C.2.** *Let $\alpha' = \alpha(1 - \epsilon')$ such that $\epsilon'$ and $v$ are real numbers satisfying $\alpha < v < 1$, $0 < \epsilon' < \min\{(1-v)/(2v), 1\}$, $0 < \epsilon' < 1/4$, $\tau \le \epsilon'\alpha \le 1/3$ and $\tau \le \epsilon/16$. Then, it holds that $|\mathcal{U}_{i+1}| \le (1 - \alpha')|\mathcal{U}_i|$ for each layer with index $i < i^*$.*

*Proof.* For the dynamic layered structure maintained by the algorithm, we use $i_{\mathrm{old}}^*$ to denote the last separating index. Then, there are mainly two cases that $i_{\mathrm{old}}^*$ is updated: (1) $i_{\mathrm{old}}^*$ is updated since a rebuild subroutine is called starting from a layer with index $i < i_{\mathrm{old}}^*$; (2) $i_{\mathrm{old}}^*$ is updated since a rebuild subroutine is called starting from the layer with index $i = i_{\mathrm{old}}^*$. We also use $n_i' = |\mathcal{U}_i^{\mathrm{OLD}}|$ to denote the size of the uncovered point set since the last time $\mathcal{U}_i$ is rebuilt.

For both cases, a key observation is that $\mathcal{U}_{i+1} \subseteq \mathcal{U}_i$ holds for the updates. Specifically, for an insertion operation that inserts a point $x$ to the dataset, $x$ will be added to both $\mathcal{U}_{i+1}$ and $\mathcal{U}_i$, and hence $\mathcal{U}_{i+1} \subseteq \mathcal{U}_i$. On the other hand, for a deletion operation that deletes a point $x$ from the dataset, if $x$ belongs to $\mathcal{U}_{i+1}$, $x$ must belong to $\mathcal{U}_i$, where a deletion operation still guarantees that $\mathcal{U}_{i+1} \subseteq \mathcal{U}_i$.

Let $i$ be an arbitrary layer index with $i < i^*$. Then, define $\kappa_i = |\mathcal{U}_{i+1}|/|\mathcal{U}_i|$ as the ratio between the uncovered sets. For an insertion operation that inserts a data point $x$ to the dataset, $x$ will be added to both $\mathcal{U}_i$ and $\mathcal{U}_{i+1}$. However, under a deletion operation that removes a data point $x$, it is possible that $x \in \mathcal{U}_i$ while $x \notin \mathcal{U}_{i+1}$. Thus, we can get that

$$\kappa_i = \frac{|\mathcal{U}_{i+1}|}{|\mathcal{U}_i|} \le \frac{n_{i+1}' + l_{\mathrm{insert}}}{n_i' + l_{\mathrm{insert}} - l_{\mathrm{delete}}},$$

where $l_{\mathrm{insert}}$ and $l_{\mathrm{delete}}$ denote the number of insertions and deletions since the last time $\mathcal{U}_{i+1}$ is reconstructed, respectively.

Then, observe that $l_{\text{insert}} + l_{\text{delete}} \leq \tau n'_{i+1} \leq \alpha n'_{i+1} = (1+\alpha)n'_{i+1} - n'_{i+1} \leq (1+\alpha)(1-\alpha)n'_i - n'_{i+1} \leq n'_i - n'_{i+1}$. By setting $A = n'_i$, $a = n'_{i+1}$, $N = l_{\text{insert}} + l_{\text{delete}}$ and using Claim C.1, we can get that

$$\kappa_i = \frac{|\mathcal{U}_{i+1}|}{|\mathcal{U}_i|} \leq \frac{n'_{i+1} + \tau n'_{i+1}}{n'_i} \leq (1-\alpha) + \tau \leq 1 - \alpha',$$

where the second to the last inequality follows from $n'_{i+1}/n'_i \leq (1-\alpha)$ (which is guaranteed by the construction of static layered structures) and the last inequality follows from $\tau \leq \epsilon'\alpha$. $\qquad\square$

**Lemma C.3.** *There are always $h = \tilde{O}(1/\epsilon)$ layers maintained by the proposed dynamic algorithm.*

*Proof.* According to Lemma C.2, since $|\mathcal{U}_1| = n$ and the sizes of uncovered sets decrease geometrically (i.e., $|\mathcal{U}_{i+1}| \leq (1-\alpha')|\mathcal{U}_i|$), it holds trivially that $i^* = \tilde{O}(1/\epsilon)$ since we require $\alpha' = \alpha(1-\epsilon')$ and $0 < \epsilon' < 1/4$.

Then, we consider the layers with indices larger than $i^*$. For an insertion operation that adds a data point $x$, there are two subcases may happen: (1) $x$ is not selected as a new representative in the $i^*$-th layer; (2) $x$ is selected as a new representative and is added to the set $S_{i^*}$ in the $i^*$-th layer. If subcase (1) happens, $x$ is directly added to the set $\mathcal{Z}$ of candidate outliers and each uncovered set $\mathcal{U}_i$ with $i \geq i^*$, where the number of layers with indices larger than $i^*$ remain the same as the last time the dynamic layered structure is updated. Note that the dynamic layered structure can only be updated starting from the a layer with index smaller than $i^*$ and $h - i^* = \tilde{O}(1/\epsilon)$. Hence, $h = \tilde{O}(1/\epsilon) + \tilde{O}(1/\epsilon) = \tilde{O}(1/\epsilon)$ holds for subcase (1).

Next, we consider that subcase (2) happens. In this subcase, the layered structure is updated starting from the $i^*$-th layer. According to the invariant (Invariant 3.2), we have $|\mathcal{U}_i| \leq (1+\tau)n'_i \leq (1+\tau)8z$, where $n'_i$ is the size of $\mathcal{U}_i$ since the last time it was reconstructed. Then, observe that each layer with indices larger than $i^*$ consists of $\epsilon z/2$ covered points at the time step the layered structures are constructed from the $i^*$-th layer, we have $h - i^* \leq \frac{(1+\tau)16z}{\epsilon z} = \tilde{O}(1/\epsilon)$. Consequently, $h = \tilde{O}(1/\epsilon) + \tilde{O}(1/\epsilon) = \tilde{O}(1/\epsilon)$ holds for subcase (2).

On the other hand, if a data point $x$ is deleted, there are also two subcases that we should consider: (1) $x$ is a representative in $S_{i^*}$; (2) $x$ is not a representative in $S_{i^*}$. If subcase (1) happens, the dynamic data structure is rebuilt from the $i^*$-th layer, where we have $h = \tilde{O}(1/\epsilon)$ using similar analysis for the insertion operations. If subcase (2) happens, $x$ is directly removed from each $\mathcal{U}_i$ and $X_i$, or the set $\mathcal{Z}$ of candidate outliers, where the number of layers with indices larger than $i^*$ remain the same as the last time the dynamic layered structure is updated. This indicates that $h = \tilde{O}(1/\epsilon)$ also holds in this subcase.

Putting all these subcases together, Lemma C.3 can be proved. $\qquad\square$

In the following, the insertion and deletion operations are discussed separately. We first consider an insertion operation, where a data point $x$ is inserted into the dataset. For each layer with indices $i < i^*$, we will show that the clustering radii of each layer can be bounded by the optimal radius of the clustering instance. Note that we fix real numbers $\upsilon$ and $\epsilon'$ such that $\alpha < \upsilon < 1$, $0 < \epsilon' < \min\{(1-\upsilon)/(2\upsilon), 1\}$ and $0 < \epsilon' < 1/4$. In the following, we use $\rho'_i = \min\{\rho \in \mathbb{R} : S' \subseteq \mathcal{U}_i, |S'| = \xi', |B(S', \mathcal{U}_i, \rho)| \geq \alpha(1+2\epsilon')|\mathcal{U}_i|\}$ to denote the minimum covering radius that covers at least a $\alpha(1+2\epsilon')$ fraction of points in $\mathcal{U}_i$ using a sample size of $\xi' = \lambda \cdot \max\{k, \log|\mathcal{U}|\}$, where $\lambda$ is a large enough constant. Recall that $\rho(S', \mathcal{U}_i) = \min\{r > 0 : |B(S', \mathcal{U}_i, r)| \geq \alpha|\mathcal{U}_i|\}$ is the minimum radius such that balls centered at points in $S'$ can cover at least an $\alpha$-fraction of points in $\mathcal{U}_i$. Denote $\rho_i^{\text{OLD}^*} = \arg\min_{S' \subset \mathcal{U}_i^{\text{OLD}} : |S'| = \xi'} \rho(S', \mathcal{U}_i^{\text{OLD}})$ as the optimal covering radius in the $i$-th layer since the last time $\mathcal{U}_i$ is reconstructed. The following lemmas show that the update-delay rule leads to covering radii that can be bounded by the optimal covering radii of dynamic data streams.

**Lemma C.4.** (Chen et al., 2018) *There exists a sufficiently large choice of $\lambda$ such that $\rho_i \leq 2\rho_i^{\text{OLD}^*}$ for each $i \in [i^* - 1]$ holds with at least constant probability.*

**Lemma C.5.** (Bhattacharya et al., 2023) *Given any two subsets $\mathcal{U}_i^{\text{OLD}}$ and $\mathcal{U}_i$ such that $|\mathcal{U}_i^{\text{OLD}} \oplus \mathcal{U}_i| \leq \epsilon'\alpha'|\mathcal{U}_i^{\text{OLD}}|$, we have $\rho_i^{\text{OLD}^*} \leq 2\rho'_i$ holds for each $i < i^*$.*

**Lemma C.6.** *With at least constant probability, it holds that $\rho_i \leq 8L^*$ for each $i < i^*$.*

*Proof.* Note that for $i \in [i^* - 1]$, we have $|\mathcal{U}_i^{\text{OLD}}| \geq 8z$. Observe that we also have $|\mathcal{U}_i| \geq (1-\tau)|\mathcal{U}_i^{\text{OLD}}|$ (for $\tau|\mathcal{U}_i^{\text{OLD}}|$ deletion operations in the worst case). Define $\mathcal{I}_i = \mathcal{I} \cap \mathcal{U}_i$ be the set of inliers in $\mathcal{U}_i$.

Similar to the analysis in static setting, we divide the optimal clusters into two subgroups based on their intersections with $\mathcal{U}_i$. Recall that $\mathcal{H}(P) = \{P_1^*, P_2^*, ..., P_k^*\}$ is the set of optimal clusters. Let $\mathcal{P}(\mathcal{U}_i) = \{P_h^* : P_h^* \in \mathcal{H}(P), P_h^* \cap \mathcal{U}_i \neq \emptyset\}$ be the

set of optimal clusters that intersect with the uncovered points in $\mathcal{U}_i$. Define $G_i = \{P_h^* : |P_h^* \cap \mathcal{U}_i| \geq |\mathcal{U}_i|/(8|\mathcal{P}(\mathcal{U}_i)|)\}$ as the set of optimal clusters containing data points that account for at least a $1/(8|\mathcal{P}(\mathcal{U}_i)|)$ fraction of the uncovered data points in $\mathcal{U}_i$ and let $M_i = \mathcal{P}(\mathcal{U}_i) \backslash G_i$. Based on the update strategy, a key observation is that $|\mathcal{U}_i| \geq 8(1 - \tau)z$. This indicates that $|\mathcal{I}_i|/|\mathcal{U}_i| \geq 1 - \frac{1}{8(1-\tau)}$ since there are only $z$ true outliers in $\mathcal{U}_i$. Then, it holds trivially that $|G_i| > 0$, as otherwise the uncovered inliers only take a fraction smaller than $1/8 < 1 - \frac{1}{8(1-\tau)}$ of the uncovered inliers (since $\tau < 1/3$), which contradicts with the fact that $|\mathcal{I}_i|/|\mathcal{U}_i| \geq 1 - \frac{1}{8(1-\tau)}$.

Next, consider an arbitrary optimal cluster $P_h^* \in \mathcal{P}(\mathcal{U}_i)$. Let $c_h \in P_h^* \cap \mathcal{U}_i$ be any uncovered data point in $P_h^*$. According to the triangle inequality, it holds trivially that the ball centered at $c_h$ with radius $2L^*$ (i.e., $B(c_h, \mathcal{U}_i, 2L^*)$) can cover all the data points in $P_h^* \cap \mathcal{U}_i$. Then, define $S_i'$ as the set of centers which contain exactly one data point from each $P_h^*$ such that $P_h^* \in G_i$. It holds that $|B(S_i', \mathcal{U}_i, 2L^*)|/|\mathcal{U}_i| \geq (1 - 1/8 - \frac{1}{8(1-\tau)}) \geq \frac{11}{16} \geq \alpha(1 + 2\epsilon')$ since we also require $\alpha < 1/3$ and $\epsilon' < 1/4$. This implies $\rho_i' \leq 2L^*$, because by definition $\rho_i^*$ is the smallest radius over all center sets with size $\xi'$ ($\xi' > |S_i'|$) such that balls of radius $\rho_i'$ can cover at least an $\alpha(1 + 2\epsilon')$-fraction of uncovered points in $\mathcal{U}_i$.

By combining with Lemma C.4 and Lemma C.5, we can conclude that with at least constant probability, it holds that $\rho_i \leq 2\rho_i^{\text{OLD}^*} \leq 4\rho_i' \leq 8L^*$. $\qquad\square$

According to Lemma C.6, the layered structures with indices $i < i^*$ can always induce a covering radius smaller than $8L^*$. During the updates, the insertion operations will not induce expansions on radius. However, for the deletion operations, alternatives may be selected to replace the representative to be deleted, which induces a 2-factor loss on the covering radius. Overall, the covering radius can be bounded by $16L^*$ for layers with indices smaller than $i^*$.

Then, we only need to consider the approximations on clustering quality for layers with indices larger than $i^*$. The following invariant shows that the Bernoulli strategy used in our proposed algorithm can maintain the same distribution as static settings with constant probability for each data insertion operation.

**Invariant C.7.** *Suppose a point $x$ is inserted at time $t$, then Insert$(x, \tau)$ and Residual-Peeling$(i^*, \mathcal{U}_{i^*} \cup \{x\}, k, z, \epsilon)$ share the same probability distribution with constant probability.*

*Proof.* For an insertion operation that inserts the data point $x$ to the dataset $P$, the proposed algorithm adapts a Bernoulli strategy to simulate the sampling process. More specific, in the $i^*$-th layer, it enumerates each representative $s \in S_{i^*}$ and replaces $s$ with $x$ using the Bernoulli$(\frac{1}{|\mathcal{U}_{i^*}|+1})$ distribution. In the following, we will analyze the probability that each data point $p' \in \mathcal{U}_{i^*} \cup \{x\}$ is selected as the representative in $S_{i^*}$.

The proof is based on an induction. We first analyze the case for $\mathcal{U}_{i^*}^{\text{OLD}} \cup \{x'\}$, where $x'$ is the first point added to $\mathcal{U}_i$ since the last rebuild. For the data point $x'$, it will replace each $s \in S_{i^*}$ with probability $\frac{1}{|\mathcal{U}_{i^*}^{\text{OLD}}|+1}$. Thus, the probability that $s$ is added to $S_{i^*}$ should be $\frac{|S_{i^*}|}{|\mathcal{U}_{i^*}^{\text{OLD}}|+1}$. For each data point $p' \in \mathcal{U}_{i^*}^{\text{OLD}}$, the probability that $p'$ is selected as the $i$-th representative should be $\frac{1}{|\mathcal{U}_{i^*}^{\text{OLD}}|}$ according to the layer construction rules. Since $p'$ is replaced by $x$ with probability $(\frac{1}{|\mathcal{U}_{i^*}|+1})$, the probability that $p'$ is selected as one of the representative is

$$\Pr(p' \in S_{i^*}) = \sum_{q=1}^{|S_{i^*}|} \frac{1}{|\mathcal{U}_{i^*}|} \cdot (1 - \frac{1}{|\mathcal{U}_{i^*}| + 1}) = \frac{|S_i^*|}{|\mathcal{U}_i^*| + 1}.$$

By combining these two cases and using an induction, Invariant C.7 can be proved. $\qquad\square$

Based on $S_{i^*}$, we can divide the data points in $\mathcal{U}_{i^*} \cap \mathcal{I}$ into different groups. Let $\mathcal{L}_{i^*} = \{P_h^* : |P_h^* \cap \mathcal{U}_{i^*}| \geq \epsilon z/(2k)\}$ and $\mathcal{M}_{i^*} = \{P_h^*, P_2^*, ..., P_k^*\} \backslash \mathcal{L}_{i^*}$ be the set of large and small optimal clusters, respectively. Similar to the static setting, our goal is to maintain a small set of samples to well represent the large optimal clusters. By combining Invariant C.7 with Lemma B.4, the following Corollary shows that $S_{i^*}$ contains at least one data point from each $P_h^* \in \mathcal{L}_{i^*}$.

**Corollary C.8.** *With constant probability, $S_{i^*}$ contains at least one data point from each large optimal cluster $P_h^* \in \mathcal{L}_{i^*}$*

For each insertion, Corollary C.8 indicates that $S_{i^*} \cap P_h^* \neq \emptyset$ holds for each $P_h^* \in \mathcal{L}_{i^*}$ with constant probability. For an insertion operation where a point $x$ is added to the dataset, observe that the total number of updates allowed is bounded

by $\tau|\mathcal{U}_{i^*}^{\mathrm{OLD}}|$ (Invariant 3.2). By setting $\tau$ as $\tau \leq \epsilon/16$, we can get that the number of updates is upper bounded by $\tau|\mathcal{U}_{i^*}^{\mathrm{OLD}}| \leq \epsilon z/2$. For each layer index $i$, let $X_i^{\mathrm{OLD}}$ be the covered set the last time it was updated. Denote $\mathcal{V} = \bigcup_{i \in [1, i^*]} S_i$ as the union of the representatives. The following lemma shows that there still exists a layer index $i'$ such that there are at most $\epsilon z$ uncovered inliers.

**Lemma C.9.** *With at least constant probability, there exists a layer index $i' \in [h]$ such that $\delta(P \backslash \mathcal{U}_{i'+1}, \mathcal{V}\}) \leq O(1)L^*$ and $|\mathcal{U}_{i'+1} \cap \mathcal{I}| \leq \epsilon z$.*

*Proof.* Since the last time the layered structures are updated using Residual-Peeling algorithm, we have $|X_i^{\mathrm{OLD}}| = \epsilon z/2$ for each $i \in [i^* + 1, h]$. Additionally, for each index $i > i^*$, the distances from covered points in $X_i$ to $S_{i^*}$ is always smaller than that for the points in $X_{i+1}$ to $S_{i^*}$. Denote $\mathcal{I}_{i^*} = \{x \in \mathcal{I} \cap \mathcal{U}_{i^*} : \delta(x, S_{i^*}) > 2L^*\}$ as the set of inliers with distances larger than $2L^*$ to the representatives in $S_{i^*}$. Based on Invariant C.7 and Corollary C.8, we have $|\mathcal{I}_{i^*}| \leq \epsilon z/2$. Let $i''$ be the largest layer index such that $\mathcal{I}_{i^*} \cap X_{i''} \neq \emptyset$ and define $\mathcal{X}' = \bigcup_{i^* \leq j \leq i''-1} X_j$ as the union of covered data points that belong to layers with indices smaller than $i''$. It holds trivially that $\delta(p, S_{i^*}) \leq 2L^*$ for each $p \in \mathcal{X}'$. This indicates that the uncovered inliers can only lie in $X_{i''}$ or the set $\mathcal{Z}$ of candidate outliers.

Since there are at most $\tau|\mathcal{U}_{i^*}^{\mathrm{OLD}}| \leq \frac{\epsilon z}{2}$ insertions or deletions before a reconstruction and points will not be inserted into $X_i$ for each $i > i^*$, we have $|X_{i''}| \leq \epsilon z$ and $|\mathcal{Z}| \leq \frac{\epsilon z}{2}$. Note that there are also $\frac{\epsilon z}{2}$ inliers belonging to small optimal clusters that cannot be covered by the representatives in $S_{i^*}$ using radius $2L^*$. Putting all these together, the uncovered inliers beyond $i''$-th layer can be bounded by $2\epsilon z$. Then, the bound for the number of uncovered inliers can be obtained by replacing $\epsilon$ with an $\epsilon'' = \frac{\epsilon}{2}$. $\square$

Then, we consider the approximation analysis for deletion operations. According to Lemma C.6, the layers with indices smaller than $i^*$ also induces an $O(1)$-approximation. For layers with indices larger than $i^*$, similar to the analysis for data insertions, the approximation can also be bounded by a constant factor of the optimal one.

## C.4. Update Time Analysis

In this subsection, we analyze the update time bounds for the proposed fully-dynamic algorithm. In the following, the update time for insertion and deletion will be analyzed separately.

### C.4.1. DATA INSERTION

Consider an insertion operation that inserts a data point $x$ to the dataset. There are two cases that may happen: (1) the dynamic data structure is updated from the $i^*$-th layer before a rebuild procedure is invoked; (2) the dynamic structure is reconstructed by invoking a rebuild procedure. If case (1) happens, according to our Bernoulli process, $x$ is added to the set $S_{i^*}$ of representatives in the $i^*$-th layer, and we show that the expected update time can be bounded by $\tilde{O}(k^2/\epsilon^3)$.

**Lemma C.10.** *Excluding the calls to Rebuild, the insert operation has expected running time of $\tilde{O}(k^2/\epsilon^3)$.*

*Proof.* Upon receiving an insertion operation, the inserted data point $x$ is first added to each layer with index $i < i^*$ and the counters are updated accordingly (steps 1-4 of Algorithm 6), which takes time $\tilde{O}(1/\epsilon)$.

Let $\mathcal{R}(i^*) = O(|\mathcal{U}_{i^*}| \cdot |S_{i^*}|/\epsilon)$ be the running time for invoking the Residual-Peeling algorithm starting from the $i^*$-th layer. Denote $\mathcal{T}(i^*)$ as the running time for executing the Bernoulli process to simulate the sampling in static settings (steps 6-20) of Algorithm 6 . According to Invariant C.7, $x$ is added to $S_{i^*}$ with probability $\frac{|S_{i^*}|}{|\mathcal{U}_{i^*}|+1}$. Then, we have

$$
\begin{aligned}
E[\mathcal{T}(i^*)] &= \frac{|S_{i^*}|}{|\mathcal{U}_{i^*}| + 1} \cdot \mathcal{R}(i^*) + (1 - \frac{|S_{i^*}|}{|\mathcal{U}_{i^*}| + 1}) \cdot \tilde{O}(1/\epsilon) \\
&= O(|S_{i^*}|^2/\epsilon) + \tilde{O}(1/\epsilon) \\
&= \tilde{O}(k^2/\epsilon^3).
\end{aligned}
$$

Putting all these together, Lemma C.10 can be proved. $\square$

C.4.2. DATA DELETION

Then, we consider the deletion operations. A key observation here is that a data point $x$ can be deleted only if it has been inserted at some earlier time step with a uniform sampling distribution. Hence, we can get the following amortized update time for deletions.

**Lemma C.11.** *Excluding the calls to Rebuild, the delete operation has expected amortized running time of $\tilde{O}(k^2/\epsilon^3)$*

*Proof.* A key observation is that $x$ can be deleted only if $x$ has been inserted. Define $Z_t(x)$ as the update time for a deletion operation at time step $t$, where $x$ is deleted from $S_{i^*}$. Recall that when $x$ is selected as a representative in $S_{i^*}$, $x$ is randomly and independently chosen from $\mathcal{U}_{i^*}^{\text{OLD}}$. Hence, $Z_t(x)$ is a random variable. Then, for a fixed $\mathcal{U}_i^*$, the update time for a deletion from the $i^*$-th layer can be bounded by $\tilde{O}(\frac{|\mathcal{U}_{i^*}|\cdot|S_i^*|}{\epsilon})$. We have

$$E[Z_t(x)|\mathcal{U}_{i^*}^{\text{OLD}}] \leq \frac{|S_{i^*}|}{|\mathcal{U}_{i^*}^{\text{OLD}}|} \cdot \tilde{O}(\frac{|\mathcal{U}_{i^*}|\cdot|S_i^*|}{\epsilon}) \leq (1+\tau)|S_{i^*}|^2/\epsilon = \tilde{O}(k^2/\epsilon^3).$$

Taking expectation of the random variable, Lemma C.11 can be proved. $\square$

C.4.3. THE REBUILD PROCEDURE

Finally, we consider the rebuild procedures. A key observation here is that the runtime for the rebuild process is dominated by the ConstructFromLayer($j$) procedure (Algorithm 9), which is indeed a greedy coverage strategy used in the static settings. According to the runtime analysis for static setting, the time for executing the reconstruction procedure from layer index $j$ can be bounded by $\tilde{O}(|\mathcal{U}_j| \cdot k/\epsilon^2)$.

**Claim C.12.** *A call to ConstructFromLayer($j$) takes time $\tilde{O}(|\mathcal{U}_j| \cdot k/\epsilon^2)$*

For a specific layer with index $i$, recall that the structures maintained in the $i$-th layer is updated only after the accumulations of updates reach a certain fraction of the covered data size (i.e., $n_i = \tau|\mathcal{U}_i|$). Intuitively, the accumulations somewhat match the time required for the the reconstruction from the $i^*$-th layer, and hence the running time of the rebuild process can be independent of the data size.

**Lemma C.13.** *A call to the rebuild takes expected $\tilde{O}(k/\epsilon^4)$ running time.*

*Proof.* To upper bound the amortized running time for the reconstructions induced by the rebuild procedure, we propose the following charging scheme. Consider a specific layer with index $i \leq i^*$, a credit of $c_1 \cdot \frac{|S_{i^*}|}{\tau\epsilon}$ is stored in the $i$-th layer when it was first constructed, where $c_1$ is a large enough constant. Then, consider the time step that the layered structure is reconstructed from the $i$-th layer by invoking a rebuild procedure. According to the update strategy, the counter $n_i$ should satisfy $n_i \geq \tau|\mathcal{U}_i^{\text{OLD}}|$, where $\mathcal{U}_i^{\text{OLD}}$ is the state of the set of uncovered points in the $i$-th layer since the last time it was reconstructed. Denote $\text{Reb}_i = \frac{c'\cdot|\mathcal{U}_i|\cdot|S_{i^*}|}{\epsilon}$ as the reconstruction time, where $c'$ is a constant. Then, we have

$$\text{Reb}_i = \frac{c' \cdot |\mathcal{U}_i| \cdot |S_{i^*}|}{\epsilon} \leq \frac{c'(1+\tau) \cdot |\mathcal{U}_i^{\text{OLD}}| \cdot |S_{i^*}|}{\epsilon} \leq \frac{c'(1+\tau) \cdot n_i \cdot |S_{i^*}|}{\tau\epsilon} \leq n_i \cdot c_1 \cdot \frac{|S_{i^*}|}{\tau\epsilon},$$

where the first inequality follows from $|\mathcal{U}_i| \leq (1+\tau)|\mathcal{U}_i^{\text{OLD}}|$, the second inequality follows from $n_i \geq \tau|\mathcal{U}_i^{\text{OLD}}|$, and the last inequality follows from $c_1$ is a large enough constant.

Since $\tau$ is roughly $\tilde{O}(\epsilon)$ and there are at most $\tilde{O}(1/\epsilon)$ layers, the amortized time for rebuild procedure can be bounded by $\tilde{O}(k/\epsilon^4)$. $\square$

Recall that the algorithm maintains $h = \tilde{O}(1/\epsilon)$ layers indexed by $i \in [h]$, and a separating index $i^* \in [h]$. For each $i \leq i^*$, the algorithm also maintains a counter $n_i$ and increments it by 1 at every insertion or deletion if the uncovered set in the $i$-th layer is updated. A reconstruction may happen for the following three cases: (1) the inserted data point $x$ is added to the set of representatives in the $i^*$-th layer; (2) the data point $x$ to be deleted belongs to the representative set in the $i^*$-th layer; (3) a rebuild procedure is invoked to construct the structured layers from the $i$-th layer for some $i \leq i^*$.

If case (1) or case (2) happens, then the expected update time for these two cases can be bounded by $\tilde{O}(k^2/\epsilon^3)$ using Lemma C.10 and Lemma C.11. According to Lemma C.13, if case (3) happens and, the time for rebuild procedure can be bounded by $\tilde{O}(k/\epsilon^4)$. Hence, the overall amortized update time for our Algorithm 2 is $\tilde{O}(k^2/\epsilon^4)$. As for the query time, it can be directly induced from Theorem 3.1 with a $\tilde{O}(k^2/\epsilon^4)$ bound. Putting all these together, Theorem 1.1 can be proved.

### C.5. Detailed Time Complexity Analysis.

Assume $k \geq \log n$, so the type-1 sample size is $\xi = O(k)$. Since $\alpha' = \Theta(\epsilon)$, the number of type-1 layers is $H_1 = O(\log n/\epsilon)$, and the number of residual layers is $O(1/\epsilon)$. The expected amortized update time consists of the cost of maintaining the layers, the expected cost of reconstructing the residual layers when the transition sample changes, and the amortized cost of counter-triggered rebuilds. Therefore the expected amortized update time is

$$O\left(\frac{k^2 \log^2(2k)}{\epsilon^3} + \frac{k \log^2 n}{\epsilon^2 \tau} + \frac{k \log n \log(2k)}{\epsilon^3 \tau}\right).$$

In particular, when $\tau = \Theta(\epsilon)$, this becomes

$$O\left(\frac{k^2 \log^2(2k)}{\epsilon^3} + \frac{k \log^2 n}{\epsilon^3} + \frac{k \log n \log(2k)}{\epsilon^4}\right).$$

For query time, the algorithm runs Center-Selection only on the weighted representative summary $\mathcal{V} = \bigcup_{i=1}^{i^*} S_i$. Under the same assumption $k \geq \log n$, its size satisfies

$$|\mathcal{V}| = O\left(\frac{k(\log n + \log(2k))}{\epsilon}\right).$$

Since Center-Selection enumerates candidate radii from pairwise distances in $\mathcal{V}$, the query time is

$$O\left(\frac{k^2(\log n + \log(2k))^2}{\epsilon^3} \log\left(\frac{2k(\log n + \log(2k))}{\epsilon}\right) \log\left(\frac{2k(\log n + \log(2k))}{\epsilon^2}\right)\right).$$

## D. Faster Algorithm for Fully-Dynamic $(k, z)$-center via Data Compression

In this section, we present a faster fully-dynamic $(k, z)$-center algorithm under mild assumptions on optimal cluster sizes. In general, if the given clustering instance satisfy that each optimal cluster has sizes comparable to the outlier budget during the whole updates, the update and query time can further be improved by a factor of $z$ using independent sampling strategies. We further complement this result with a lower bound showing that, in the general metric space query model, any algorithm with $O(1)$-approximation and $O(z)$ outliers discarded must incur $\Omega(k^2/z)$ update time.

### D.1. A Faster Fully-Dynamic $(k, z)$-center Algorithm

The intuitive idea behind our faster fully-dynamic $(k, z)$-center algorithm is to maintain an unweighted "weak coreset" during the updates. Roughly speaking, a weak coreset is a subset of the original data, where executing any $(k, z)$-center routine on the weak coreset only induces constant approximation loss while guaranteeing the number of discarded outliers.

We start by considering the static settings. Given a $(k, z)$-center instance $(P, k, z)$, the following lemma shows that by randomly sampling each point with independent probability $\frac{4 \ln(2k)}{\epsilon z}$ from $P$, a weak coreset with constant approximation loss can be obtained.

**Lemma 4.1.** *Let $(P, k, z)$ be a $k$-center with outliers instance, and let $S \subseteq P$ be a set obtained by sampling each data point independently with probability $p = \frac{4 \ln(2k)}{\epsilon z}$. Define a new instance $(S, k, \hat{z})$ with $\hat{z} = \frac{4 \ln(2k)}{\epsilon \eta}$ for some parameter $0 < \eta < 1$ such that $\epsilon > \frac{2}{3\eta}$. Under the assumption that each optimal cluster in $P$ has size at least $3\epsilon z$, if an algorithm $\mathcal{A}$ returns a $\zeta$-approximate solution with $z$ outliers discarded, a $(\zeta + 2)$-approximation on $P$ with $z$ outliers discarded can be achieved with probability $\Omega(1 - \eta)$ by executing $\mathcal{A}$ on $S$ with $\hat{z} = \frac{4 \ln(2k)}{\epsilon \eta}$.*

*Proof.* Let $\mathcal{P}(\mathcal{H}) = \{P_1^*, P_2^*, ..., P_k^*\}$ be the set of the optimal clusters. Denote $S$ as the sampled subset obtained. The proof strategy is to show that the independent sampling strategy can guarantee that the optimal clusters still take a certain

fraction of the sampled points, where at least one data point from these large optimal clusters should be covered by a $(k, z)$-center routine. We will show that the following two properties hold for optimal clusters in $\mathcal{P}(\mathcal{H})$ when each optimal cluster $P_h^*$ satisfies that $|P_h^*| \geq 3\epsilon z$.

(1) With constant probability, each optimal cluster $P_h^* \in \mathcal{P}(\mathcal{H})$ satisfies that $P_h^* \cap S \neq \emptyset$.

(2) With constant probability, each optimal cluster $P_h^* \in \mathcal{P}(\mathcal{H})$ satisfies that $|P_h^* \cap S| \in (1 \pm 1/2)|P_h^*| \cdot p$.

We first show that property (1) stated can hold with constant probability. For an arbitrary optimal cluster $P_h^*$, recall that $p = \frac{4\ln(2k)}{\epsilon z}$ is the independent sampling probability. According to the assumption, we have $|P_h^*| \geq 3\epsilon z$. Hence, the probability that at least one point is sampled should be bounded by

$$\begin{aligned} \mathrm{Pr}(S \cap P_h^* \neq \emptyset) &= 1 - (1-p)^{|P_h^*|} \\ &= 1 - e^{|P_h^*|\ln(1-p)} \\ &\geq 1 - e^{-|P_h^*| \cdot p} \\ &\geq 1 - 1/k, \end{aligned}$$

where the second to the last inequality follows from $\ln(1+x) \leq x$ and the last inequality follows from $|P_h^*| \geq 3\epsilon z$.

Hence, by independent sampling each data point with probability $p = \frac{4\ln(2k)}{\epsilon z}$, with at least constant probability, it holds that $P_h^* \cap S \neq \emptyset$ by taking a union bound on the success probability. Then, property (1) stated can be achieved.

As for property (2), this property holds via Chernoff Bounds. Define $S$ as the set of samples obtained through the independent sampling process. For each optimal cluster $P_h^*$, let $v_h = |P_h^*|$. We define random variables $\{y_1, ..., y_{v_h}\}$. For each $1 \leq i \leq v_h$, $y_i = 1$ if the $i$-th element in $P_h^*$ is added to $S$ via independent sampling. Otherwise, we set $y_i = 0$. Let $Y = \sum_{i=1}^{v_h} y_i$. We have $E[y_i] = p$ and $E[Y] = p \cdot |P_h^*|$. By setting parameter $\delta = 1/2$, the Chernoff Bound implies that

$$\mathrm{Pr}(Y \in (1 \pm \delta)E[Y]) \geq 1 - 2e^{-\frac{E[Y]\delta^2}{3}} \geq 1 - 2e^{-\frac{p \cdot |P_h^*|}{12}} \geq 1 - 1/k,$$

where the last inequality follows from $|P_h^*| \geq 3\epsilon z$ and $p = \frac{4\ln(2k)}{\epsilon z}$.

By taking a union bound on success probability, property (2) stated can hold with at least constant probability for all the optimal clusters. As for the outliers, since there are $z$ true outliers, the expected number of outliers in $S$ can be bounded by $E[|\mathcal{Z}^* \cap S|] = z \cdot p$. By applying a Markov's Inequality, for a success probability $\eta$, we can get that

$$\mathrm{Pr}\left(|\mathcal{Z}^* \cap S| \leq \frac{z \cdot p}{\eta}\right) \geq 1 - \eta.$$

Since we define $\hat{z} = \frac{4\ln(2k)}{\epsilon\eta}$, it holds that $|\mathcal{Z}^* \cap S| \leq \frac{z \cdot p}{\eta} \leq \hat{z}$. Moreover, base on property (2), we have

$$\begin{aligned} |S \cap P_h^*| &\geq \frac{1}{2}|P_h^*| \cdot p \\ &\geq \frac{3\epsilon z}{2} \cdot \frac{4\ln(2k)}{\epsilon z} \\ &\geq 6\ln(2k). \end{aligned}$$

By setting $\epsilon$ as $\epsilon > \frac{2}{3\eta}$, we have $|S \cap P_h^*| > \hat{z}$. Then, let $\mathcal{C}$ be the set of the $k$ centers returned by executing an existing $(k, z)$-center routine with approximation ratio $\zeta$ on the sampled set $S$. Define $\mathcal{C}'$ as the set of centers with size at most $k$ consisting of points in $S$ such that at least one point from each optimal cluster can be found in $\mathcal{C}'$. This induces a feasible solution such that the number of outliers discarded can be chosen from the true outliers that intersect with the sample set $S$ (i.e., $\mathcal{Z}^* \cap S$), where at most $\hat{z}$ points are discarded as outliers. Based on the center set $\mathcal{C}$, let $\mathcal{F}(\mathcal{C})$ be the set of the furthest $\hat{z}$ points in $S$ to $\mathcal{C}$. According to the properties stated for optimal cluster, $S \backslash \mathcal{F}(\mathcal{C})$ must contain at least one data from each

optimal cluster $P_h^*$. Otherwise, there are more than $\hat{z}$ outliers discarded by the algorithm, which does not induce a feasible clustering solution. This implies that a $(\zeta + 2)$-approximation can be obtained by executing a $\zeta$-approximation $(k, z)$-center routine on $S$ using triangle inequality. □

By leveraging the properties stated in Lemma 4.1, our algorithms for accelerating the fully-dynamic $(k, z)$-center problem is to perform an independent sampling step before any insertion operations. The formal algorithm is presented in Algorithm 10.

---

**Algorithm 10** FasterDynamicClustering

**Input:** A sequence $\mathcal{S} = \{o_1, o_2, ..., o_i\}$ of update operations, an initial dataset $P$, parameters $k, z, \epsilon, \alpha, \tau, \eta$.

1: Fix a sampling probability $p = \frac{4\ln(2k)}{\epsilon z}$.
2: Initialize a sample set $S$ by sampling each point in $P$ independently with probability $p$, and set $\hat{z} = \frac{4\ln(2k)}{\epsilon\eta}$.
3: Construct a data structure $\mathcal{D}'$ by calling the Layered-Sampling$(S, k, z, \alpha, \epsilon)$ algorithm, initialize $n_i = 0$ for each $i \in [i^*]$, and set $\mathcal{Z} = \emptyset$, where $i^*$ is the separating index for $\mathcal{D}'$.
4: **if** $o_i$ is to insert a point $x$ **then**
5:     Sample Bernoulli $I$, s.t. $\mathbf{Pr}[I = 1] = p$.
6:     **if** $I = 1$ **then**
7:         Insert $x$ to the current dataset.
8:         Insert $x$ into the set $S$ by executing the fully-dynamic algorithm (Algorithm 2) on $S$, where the number of outliers discarded is set as $\hat{z}$.
9:     **else**
10:         CONTINUE;
11:     **end if**
12: **end if**
13: **if** $o_i$ is to delete a point $x$ **then**
14:     Delete $x$ from the current dataset.
15:     **if** $x \in S$ **then**
16:         Delete $x$ from $S$ executing the fully-dynamic algorithm (Algorithm 2) on $S$, where the number of outliers discarded is set as $\hat{z}$.
17:     **else**
18:         CONTINUE;
19:     **end if**
20: **end if**
21: For a specific query, call the Center-Selection Algorithm (Algorithm 4) to obtain a solution by setting the input $z$ as $z - |\mathcal{Z}|$, where $\mathcal{Z}$ is the set of candidate outliers maintained by $\mathcal{D}'$.

---

Upon each insertion, a Bernoulli sampling step (step 5 of Algorithm 10) is performed to simulate that $x$ is sampled to form the weak coreset $S$. Since the sampling probability is independent of the data size, the insertion and deletion operations on the points are independent of each other, while guaranteeing the probability distributions for constructing $S$. By combining with Lemma 4.1, Theorem 1.2 holds trivially.

### D.2. Lower Bounds on the Update Time

In this subsection, we provide a lower bound for the update time of the fully-dynamic $(k, z)$-center problem. In the static setting, Grunau and Rozhoň (Grunau & Rozhoň, 2022) proves the following lower bound on the running time.

**Theorem D.1.** (Grunau & Rozhoň, 2022) *Any randomized algorithm for the $k$-means/$k$-median/$k$-center problem with outliers in the setting $k \geq C$, $z \geq Ck \log k$, and $n \geq Cz$ for an absolute constant $C$ that with probability at least $0.5$ gives an $O(1)$-approximation for these problems with $O(z)$ outliers discarded in the general metric space query model, needs $\Omega(nk^2/z)$ queries.*

At a high level, the theorem is proved by constructing a distribution over instances in the general metric space query model such that no deterministic algorithm can succeed with probability exceeding $0.5$ under this distribution. Hence, the stated randomized lower bound then follows by Yao's minimax principle (Yao, 1977).

To prove a lower bound for the update time, our strategy is to transfer this static lower bound to the dynamic setting. A key observation is that the query time has lower bound of $\Omega(k)$ since there are $k$ clusters. Consider an incremental $k$-center data structure with amortized update time $u(n, k, z)$ and query time $q(n, k, z)$. By inserting the $n$ points of one at a time and then raise a query when all the points have been inserted, the dynamic algorithm can be modified in this way to obtain an approximate solution in static settings with an overall running time of $n \cdot u(n, k, z) + q(n, k, z)$. According to Theorem D.1, we have $n \cdot u(n, k, z) + q(n, k, z) = \Omega(nk^2/z)$ in the the general metric space query model.

Then, since a fully-dynamic $(k, z)$-center algorithm should have $\Omega(k)$ query time, an update time lower bound of $\Omega(k^2/z)$ can be obtained, which is stated as following.

**Theorem 1.3** *Any randomized algorithm for the fully-dynamic $(k, z)$-center problem in the setting $k \geq C$, $z \geq Ck \log k$, and $n \geq Cz$ for an absolute constant $C$ that with probability at least 0.5 gives an $O(1)$ approximation with $O(z)$ outliers discarded in the general metric space query model, requires $\Omega(k^2/z)$ update time.*

