# OpenReview forum: "New Algorithms for Fully-Dynamic k-center with Outliers"
_ICML.cc/2026/Conference — ICML 2026 regular_

### Official Review · Reviewer_KZJ8 · 2026-03-07

**Soundness:** 3
**Presentation:** 2
**Significance:** 3
**Originality:** 2
**Overall Recommendation:** 5
**Confidence:** 4

**Summary:**

The paper studies the fully dynamic $(k,z)$-center problem in general metric spaces, where points may be inserted or deleted and up to $z$ points can be treated as outliers. The goal is to maintain a constant-factor approximation while supporting efficient update and query operations. Existing approaches typically rely on radius guessing based on the metric aspect ratio $\Delta$, which introduces an $O(\log \Delta)$ overhead in either update or query time and can become significant when $\Delta$ is large. The authors propose a layered-sampling framework that removes the dependence on the aspect ratio $\Delta$ and does not require prior bounds on pairwise distances. The algorithm maintains a sequence of layered representative structures constructed via a greedy coverage strategy, where each layer selects representatives from currently uncovered points and covers nearby points. Queries are answered by constructing weighted summaries from these representatives and running a $(k,z)$-center routine on the summaries. A preprocessing step merges nearby representatives to ensure a separation property, which allows the algorithm to identify an approximate optimal radius through distances among representatives rather than through aspect-ratio–dependent radius guessing. The resulting randomized algorithm returns an $O(1)$-approximate solution with $(1+\varepsilon)z$ outliers and achieves amortized update and query time $\tilde{O}(k^2/\varepsilon^4)$. Under the additional assumption that each optimal cluster has size $\Omega(z)$, the authors further present a faster algorithm using data compression through independent sampling, which reduces the effective data size. This algorithm achieves $\tilde{O}(k^2/z)$ update and query time while discarding $O(z)$ outliers. Finally, the paper proves a matching lower bound of $\Omega(k^2/z)$ on the update time in the general metric query model, showing that the improved algorithm is near-optimal under this model.

**Compliance With Llm Reviewing Policy:**

Affirmed.

**Ethical Review Concerns:**

None.

**Final Justification:**

I was already in the favour of this paper. I do not have anything more to add.

**Key Questions For Authors:**

Does there algorithm work in MPC or other model?

**Limitations:**

The paper does not explicitly discuss the limitations of the proposed approach. In particular, it would be helpful for the authors to clarify whether the techniques extend to related clustering objectives such as $k$-median or $k$-means, and to discuss potential challenges or limitations in applying the framework to these settings.

**Strengths And Weaknesses:**

Soundness: All the poofs that I could verify seem correct. The algorithms are clearly specified and the main theoretical guarantees are supported by analysis.

Presentation: The paper provides extensive discussion of the high-level intuition behind the proposed layered-sampling framework in the main text. However, some parts of this discussion appear somewhat repetitive, while many technical details and analyses are deferred to the appendix. The presentation could be improved by reducing redundancy in the high-level explanations and including more of the key technical insights or proof ideas in the main text.

Significance: The paper tackles an important problem in dynamic clustering and proposes algorithms that eliminate the dependence on the metric aspect ratio while achieving efficient update and query time.

Originality: The paper proposes a layered-sampling approach to handle the fully dynamic $(k,z)$-center problem without relying on aspect-ratio–dependent radius guessing. While the problem and setting have been studied previously, the proposed framework and its analysis  provides some novel algorithmic ideas. However, layered-sampling approach has been used previously in literature.

---

> ### Author Rebuttal · Authors · 2026-03-31
>
> We thank the reviewer for the positive rating and constructive feedbacks. Below, we address the concerns.
>
> **Question 1: Does the algorithm work in MPC or other model?**
>
> Thank you for raising this important question. Our current results are proved only in the fully dynamic model. However, the framework has the potential to extend to the MPC setting to remove aspect-ratio dependence from local running time, communication, and local memory.
>
> Our method is built from two main ingredients: independent sampling and ball division. The sampling component is based on random/uniform sampling and is naturally parallelizable, while the main obstacle is the ball-division step. In geometric settings, a possible route is to combine our framework with hashing-based tools such as consistent hashing or LSH [1]-[3] to approximately implement this step, which may enable an MPC version with only a small additional approximation loss.
>
> Similarly, the framework has the potential to extend to k-median/k-means objectives. Prior work already shows that layered sampling is feasible for these objectives in static settings [4]. This suggests that the framework can generally be extended to other k-clustering objectives, although extending it to other objectives would require new analysis.
>
> We will highlight this as an interesting direction for future work in the revised version. We will also include a more detailed discussion on whether our techniques can be extended to related clustering objectives such as k-median and k-means, as well as the potential challenges and limitations in adapting the framework to these settings.
>
> [1] Artur Czumaj, et al. Fully-Scalable MPC algorithms for clustering in high dimension. In Proceedings of the 51st International Colloquium on Automata, Languages, and Programming (ICALP), 2024.
>
> [2] Sam Coy, Artur Czumaj, and Gopinath Mishra. On parallel k-center clustering. In Proceedings of the 35th ACM Symposium on Parallelism in Algorithms and Architectures (SPAA), pp. 65-75, 2023.
>
> [3] MohammadHossein Bateni, Hossein Esfandiari, Manuela Fischer, and Vahab Mirrokni. Extreme k-center clustering. In Proceedings of the AAAI Conference on Artificial Intelligence (AAAI), pp. 3941-3949, 2021.
>
> [4] Jiecao Chen, et al. A practical algorithm for distributed clustering and outlier detection. In Proceedings of the 35th International Conference on Neural Information Processing Systems (NeurIPS), pp.2248–2256, 2018

---

> > ### Author Rebuttal · Reviewer_KZJ8 · 2026-04-01
> >
> > I will keep my earlier score. While I raised some concerns, they do not change my overall assessment. The authors have indicated that they will address them in a future version.

---

### Official Review · Reviewer_j8xF · 2026-03-08

**Soundness:** 3
**Presentation:** 2
**Significance:** 2
**Originality:** 2
**Overall Recommendation:** 4
**Confidence:** 3

**Summary:**

This paper studies the fully dynamic $k$-center clustering problem with outliers. Main goal of this paper is to design a dynamic clustering algorithm with sublinear update and query time and meanwhile to avoid dependence on the aspect ratio $\Delta$ that is commonly present in prior algorithms. The proposed layered-sampling algorithm achieves constant approximation with at most $(1+\epsilon) z$ outliers being removed.

**Compliance With Llm Reviewing Policy:**

Affirmed.

**Final Justification:**

The rebuttal consolidated my concerns about the weaknesses (relevance of the k-center method to the ML community and empirical advantage of the layer-sampling algorithm). I updated my score from 3 to 4, reflecting my positive recommendation after rebuttal.

**Key Questions For Authors:**

See the Weakness points above.

A minor typo.
- Page 5, line 262: Roadamap -> Roadmap

**Limitations:**

There is no discussion of limitations in this paper.

**Strengths And Weaknesses:**

Strength

- This is the first fully dynamic and aspect-ratio independent $(k, z)$-center method.
- The layer-sampling method matches a lower bound of update time in the general metric space query model.
- Theoretical contributions appears to be solid.

Weakness

- This paper is highly technical. The current form of writing and structure are very difficult to understand and extract its core insight. The introduction on prior works in literature has multiple entangled lines and it is hard to compare the derived results with those in terms of main contribution and limitation, where the latter is not discussed in this paper.
- I am not sure how the $k$-center clustering is relevant to the ML community, where the $k$-means clustering is far more popular. I understand that there are scenarios that worst-case distance is more relevant than averaged squared distance. However, I am not convince based on the current form that the proposed layer-sampling method for solving the $k$-center clustering with outlier has broad enough implications in practical ML. Therefore, I think this paper is perhaps more suitable for a pure algorithmic or learning-theoretical venue.
- Consistent with the weakness noted above, the paper does not provide numerical experiments demonstrating the empirical advantages of the proposed layered-sampling method in the streaming setting compared with existing algorithms.

---

> ### Author Rebuttal · Authors · 2026-03-31
>
> We thank the reviewer for the constructive feedbacks and thoughtful comments. Below, we address the concerns.
>
> **Weakness 1: Regarding the clarity of the presentation and the discussion of the improvements and limitations of prior work**
>
> Response: Thank you for the constructive feedback. We agree that the current presentation can be improved, especially in making the comparison with prior work clearer and the limitations more explicit. Below, we briefly summarize the main improvements and current limitations.
>
> Our main improvement arises in high-aspect-ratio scenarios, where the explicit $\log\Delta$ dependence in prior fully dynamic methods can dominate the overhead. Our method also does not require prior bounds on pairwise distances, making it applicable when the metric scale is unknown or varies over time. The tradeoff is a larger approximation ratio and additional polylogarithmic dependence in the time bounds. When $\Delta$ is small, the benefit may be less significant.
>
> In the revision, we will discuss the improvements and limitations explicitly, and reorganize the introduction and related work for clearer comparison.
>
> **Weakness 2: Regarding the relevance of the k-center and the broader practical implications of layered sampling for ML**
>
> Response: Thank you for raising this point. The k-center problem belongs to the same family of center-based clustering objectives as k-median and k-means, differing mainly in how assignment costs are defined. Thus, k-center captures a complementary notion of clustering quality, which is also relevant in ML.
>
> Additionally, both k-center and k-center with outliers have been extensively studied in ML community, including core-set based batch active learning [1] and settings with noisy [2] or fully dynamic data [3]. In such regimes, worst-case coverage and robustness to outliers are natural requirements, making fully dynamic k-center with outliers relevant to ML rather than purely algorithmic.
>
> Methodologically, our contribution is not only a new result for k-center, but also an extension of layered sampling to robust fully dynamic setting. More broadly, layered sampling can serve as a useful tool for accelerating practical ML algorithms on large-scale, noisy, and evolving data. In particular, it has the potential to be applied to robust dynamic k-median/k-means clustering and other $\Delta$-dependent tasks such as approximate nearest neighbor search.
>
> In the revised version, we will include a more explicit discussion on these points.
>
> [1] Ozan Sener, and Silvio Savarese. Active Learning for Convolutional Neural Networks: A Core-Set Approach. ICLR 2018.
>
> [2] Junyu Huang, et al. Fast algorithms for distributed k-clustering with outliers. ICML 2023
>
> [3] Sayan Bhattacharya, et al. Almost Optimal Fully Dynamic k-center Clustering with Recourse. ICML 2025.
>
> **Weakness 3: Regarding the lack of empirical evaluation compared with previous methods**
>
> Response: Thank you for pointing this out. In line with prior work on fully dynamic clustering problems, our original submission mainly focused on theoretical improvements. In addition, since public implementations of the most relevant baselines are unavailable, a fair comparison requires implementing them from their descriptions.
>
> Since fully dynamic setting is more general than streaming, following the prior work [4], we address this concern by adding  experiments on synthetic data under sliding-window model. In this setting, each point inserted at time t remains active for W steps and is deleted at time t+W, so each step can involve both insertions and deletions. Queries are issued every *query\_every* steps. We compare our method with (14+ϵ)-approximation Two-Stage method of Chan et al., (4+ϵ)-approximation Sampling-Greedy method of Biabani et al., and a static Greedy baseline [5]. Table 1 reports the ratio of average clustering cost to the static baseline, total update time, and total query time, where our method achieves the best total update time with comparable clustering quality to prior methods.
>
> [4] Sayan Bhattacharya, et al. Fully Dynamic k-Clustering in $\tilde{O}(k)$ Update Time. NeurIPS 2023.
>
> [5] Aditya Bhaskara, et al. Greedy sampling for approximate clustering in the presence of outliers. NeurIPS 2019.
>
> *Table 1: Comparison results on synthetic datasets, where k=10, z =2%n, parameter $\alpha$=0.6, $\epsilon$=0.5, $\tau$=1.0*
>
> |W=500,n=2000,query\_every=100||||
> |:-----------------------------------:|:---------------------------:|:------------------------:|:------------------------:|
> |Method|Ratio-to-Baseline(Cost)|Total_Update\_Time(s)|Total\_Query_Time(s)|
> |Two-Stage|1.3481|13.0885|0.1372|
> |Sampling-Greedy|**1.1236**|126.18|**0.0008**|
> |Ours|1.1408|**0.7866**|0.6873|
> |||||
> |**W=3000,n=10000,query\_every=500**||||
> |Method|Ratio-to-Baseline(Cost)|Total_Update_Time(s)|Total_Query_Time(s)|
> |Two-Stage|1.4363|70.039|0.1129|
> |Sampling-Greedy|**1.0844**|492.775|**0.0008**|
> |Ours|1.1246|**3.5125**|0.7517|
> |||||

---

> > ### Author Rebuttal · Reviewer_j8xF · 2026-04-02
> >
> > Thanks for your detailed clarification and additional numeric examples. I see a better connection to other centroid-based clustering methods and the empirical advantage of the layer-sampling method in terms of total update time. I will correspondingly raise my score.

---

### Official Review · Reviewer_tWpQ · 2026-03-10

**Soundness:** 4
**Presentation:** 3
**Significance:** 2
**Originality:** 4
**Overall Recommendation:** 4
**Confidence:** 4

**Summary:**

This paper introduces a new algorithm for the k-center problem with outliers in the fully dynamic setting, in general metric spaces. The core contribution is removing the dependence on the aspect ratio of the metric space, i.e., the ratio between the diameter and the minimum pairwise distance. The algorithm is randomized, achieves a $24(1+\varepsilon)$ bi-criteria approximation with $z \cdot (1+\varepsilon)$ outliers and has $\tilde{O}(k^2 / \varepsilon^4)$ update and query times. The main idea is to keep a layered data structure where each layer maintains some centers that cover a constant fraction of the points left uncovered by the previous layer, where each layer induces a radius based on the points it cover. This removes the need for guessing radii. In the fully dynamic setting, the data structure is updated lazily to keep update times tame. Moreover, under the assumption that no optimal cluster is too small, the authors present improved results that yield an $O(k^2/z)$ update time. Finally, a novel lower bound for updates times for k-center with outliers with is also presented, matching the previous result.

**Compliance With Llm Reviewing Policy:**

Affirmed.

**Final Justification:**

I still think that the removal of the $\log \Delta$ factor can only partially make up for the worsening of the approximation factor and the update time dependency on the other parameters. My general assessment remains positive.

**Key Questions For Authors:**

See weaknesses.

**Limitations:**

I would suggest the authors to include some limitations and directions for future work.

**Strengths And Weaknesses:**

Strengths:
- the paper is really well written. It first gives good intuitions on the stages of the algorithm before digging into the details. I particularly like the fact that it first introduces the static algorithm and then explains how to make it dynamic.
- the paper does a very thorough overview of the related work and gives context to the present contribution (but see some minor remarks below).
- the paper somewhat under-sells the contribution, as (to the best of my knowledge) this is the first fully dynamic algorithm without the aspect ratio dependence even for the k-center problem without outliers (by setting z to 0). I think it would be fair to have at least a remark about this.
- the technical contributions are original and they seem correct to me, although I was unable to check all the details.

Weaknesses:

The following weaknesses are quite minor (I still believe that the algorithm is a good contribution and it is technically very complex), but they should still be discussed.
- the 24+eps approximation is a steep price to pay, especially compared to the 14+eps approximation given by Chan et al. , which has lower dependency on k and $\varepsilon$ in the update and query times.
- the comparisons in Table 1 are a bit unfair, as the $\tilde{O}$ notations hides logarithmic factors only for the present paper, and not for the others. You should re-introduce the $\log n$ term either in the table, in the theorem, or in a discussion. I find it confusing that the $\log n$ term is hidden even in the proofs in the appendix (e.g., Lemma C.3).
- In fact, I think that in most scenarios $\log \Delta$ would be preferable compared to the $O(\log n)$ term you obtain (and I think an additional $\log k$ term from Residual-Peeling, but I’m not sure) .

Minor remarks:
- the LP-based algorithm from [1] yields a 2+eps approximation in doubling metrics, which is the SOTA with respect to approximation ratios for fully dynamic k-center with outliers.
- In Lemma C.10, you are missing a tilde in the statement.
- The lower bound of $\Omega(z)$ update time for exactly z outliers has been established in the journal version [2], not Chan et al. 2022.

.

- [1] Pellizzoni et al. Fully Dynamic Clustering and Diversity Maximization in Doubling Metrics. ACM TKDD (2025).
- [2] Chan et al. Fully Dynamic k-Center Clustering with Outliers. Algorithmica (2024).

---

> ### Author Rebuttal · Authors · 2026-03-31
>
> We thank the reviewer for the positive rating and constructive feedbacks. Below, we address the concerns.
>
> **Weakness 1: The $24+\epsilon$ approximation is a steep price to pay, especially compared to the $14+\epsilon$ approximation given by Chan et al., which has lower dependency on k and $\epsilon$ in the update and query times.**
>
> Response: Thank you for pointing this out. We agree with the reviewer that our worst-case approximation guarantee is weaker than the prior result, such as Chan et al. Nevertheless, our result gives the first fully dynamic algorithm whose update and query times are independent of the aspect ratio $\Delta$, while also avoiding any prior assumptions about the lower or upper bounds on pairwise distances. We believe that this modest increase in the approximation ratio, together with the additional polylogarithmic factors, is acceptable given that it eliminates the explicit $\Delta$-dependence from both update and query time, especially in high-aspect-ratio settings. In such regimes, the explicit $\log\Delta$ term in previous methods can become the dominant overhead and may far exceed the polynomial dependence on k and 1/ϵ. Improving the approximation ratio while still preserving aspect-ratio-independent update and query time is an interesting direction for future work.
>
> **Weakness 2: The comparisons in Table 1 are a bit unfair, as the $\tilde{O}$ notations hides logarithmic factors only for the present paper, and not for the others. You should re-introduce the $\log n$ term either in the table, in the theorem, or in a discussion. I find it confusing that the $\log n$ term is hidden even in the proofs in the appendix (e.g., Lemma C.3).**
>
> Response: Thank you for the constructive feedback. We agree with the reviewer that using $\tilde{O}$ notation only for our results can make the comparison appear unfair. Our intention was to simplify the presentation and highlight the technical contributions, namely the removal of the $\Delta$-dependence in update and query times while suppressing lower-order polylogarithmic factors. In the revised version, we will clarify this point by explicitly presenting the suppressed terms and the corresponding polylogarithmic factors in Table 1, Theorem 1.1, and Theorem 1.2, along with an added remark below.
>
> **Weakness 3: In fact, I think that in most scenarios $\log\Delta$ would be preferable compared to the $O(\log n)$ term you obtain (and I think an additional $\log k$ term from Residual-Peeling, but I’m not sure).**
>
> Response: Thank you for pointing this out. We agree with the reviewer that when the aspect ratio $\Delta$ is polynomially bounded in the data size, the $O(\log\Delta)$ dependence in prior work can be comparable to the $O(\log n)$ and additional $O(\log k)$ factors in our bounds. However, as pointed out in prior work [1], $\Delta$ can be much larger than polynomial in the input size (as large as $2^{n^{o(1)}}$), and in the worst case may even be arbitrarily large [2]-[3]. Such large aspect ratios can also arise in practice, for example in high-precision sensing, GIS data, or datasets with many near-duplicate points, and this can naturally occur in fully dynamic settings as the metric scale changes over time. In these regimes, removing the explicit dependence on $\Delta$ yields a clear advantage.
>
> We will include detailed discussions of such comparisons in the revised version.
>
> **Minor Remark 1: the LP-based algorithm from [1] yields a $2+\epsilon$ approximation in doubling metrics, which is the SOTA with respect to approximation ratios for fully dynamic k-center with outliers.**
>
> Response: Thank you for this helpful correction. In [1], the authors present both a $(3+\epsilon)$-approximation combinatorial algorithm and a stronger $(2+\epsilon)$-approximation LP-based algorithm. Our previous discussion only mentioned the former, and we apologize for the resulting confusion. In the revised version, we will update the related-work section to properly acknowledge the $(2+\epsilon)$-approximation SOTA result in doubling metrics.
>
> **Minor remark 2: In Lemma C.10, you are missing a tilde in the statement.**
>
> Response: Thank you for the constructive feedback. We will correct Lemma C.10 so that the statement is consistent.
>
> **Minor remark 3: The lower bound of $\Omega(z)$ update time for exactly z outliers has been established in the journal version [2], not Chan et al. 2022.**
>
> Response: Thank you for the correction. We will update the citation and attribution accordingly in the revised version.
>
> [1] Vincent Cohen-Addad, Vahab Mirrokni, and Peilin Zhong. Massively parallel k-means clustering for perturbation resilient instances. In Proceedings of the International Conference on Machine Learning, 2022.
>
> [2] Huy Lê Nguyen, et al. Fair range k-center. arXiv preprint arXiv:2207.11337 (2022).
>
> [3] Robi Bhattacharjee, and Michal Moshkovitz. No-substitution k-means clustering with adversarial order. In Proceedings of  Algorithmic Learning Theory, 2021.

---

> > ### Author Rebuttal · Reviewer_tWpQ · 2026-04-02
> >
> > Thank you for addressing W2 and the minor remarks.
> > I still think that the removal of $\log \Delta$ factor can only partially make up for the worsening of the approximation factor and the update time dependency on the other parameters, and I agree that addressing this is an interesting direction for future work.
> >
> > I will keep my positive score.

---

### Official Review · Reviewer_YVKc · 2026-03-13

**Soundness:** 3
**Presentation:** 3
**Significance:** 3
**Originality:** 3
**Overall Recommendation:** 5
**Confidence:** 3

**Summary:**

This work explores the fully dynamic k-center problem with outliers. The authors present two main results:

The first result improves update time by making it independent of $\Delta$, though this comes at the cost of an increased approximation guarantee (and query time). Furthermore, the authors improve these results by assuming that the size of each cluster is linear with respect to $z$.

**Compliance With Llm Reviewing Policy:**

Affirmed.

**Final Justification:**

Strong and well presented publication with good contribution to the field.

**Key Questions For Authors:**

There are multiple algorithms that bound the value of $\Delta$ in the static setting. Are there any such algorithms in the fully dynamic setting?

**Limitations:**

yes

**Strengths And Weaknesses:**

Strengths:

- Overall, the paper is well-written and easy to read. The results are clearly stated with sufficient detail, and the paper successfully delivers on its claims.

- Removing the dependency on $\Delta$ is important and is of theoretical interest.

- The secondary result is also quite compelling, and its significance is further highlighted by the hardness result.

- Both the $k$-center problem and the fully dynamic setting are well-known and significant areas of research in this field.

Weaknesses:

- The increase in the approximation ratio is significant.

- The algorithm is bicriteria (similar to most state-of-the-art algorithms).

- The assumption of the size in Theorem 1.2 and 1.3 are different.

---

> ### Author Rebuttal · Authors · 2026-03-31
>
> We thank the reviewer for the positive rating and constructive feedbacks. Below, we address the concerns.
>
> **Question 1: There are multiple algorithms that bound the value of $\Delta$ in the static setting. Are there any such algorithms in the fully dynamic setting?**
>
> Response: Thank you for raising this important question. To the best of our knowledge, we are not aware of any existing algorithm that can dynamically bound the value of $\Delta$ in the fully dynamic setting.
>
> In the static setting, methods that can bound $\Delta$ by poly(n) typically rely on modifying the clustering instance itself, rather than directly maintaining such a bound on $\Delta$ [1]-[3]. At a high level, these approaches start by obtaining a crude bound on the optimal clustering cost, and then transform the instance so that nearby points can be grouped locally, while assignments over distances much larger than this bound become irrelevant. This makes it possible to control both the diameter and the smallest relevant nonzero distance relative to the cost bound, resulting in a transformed instance whose aspect ratio is bounded by poly(n).
>
> However, such instance transformations are much harder to support in the fully dynamic setting. In particular, even maintaining a crude upper bound on the optimal clustering cost under insertions and deletions is already nontrivial, especially in the presence of outliers. Moreover, dynamically updating the transformed instance would itself incur substantial overhead. Therefore, while the static setting admits techniques that can effectively bound the aspect ratio through instance modification, no analogous approach is currently known in the fully dynamic setting. Designing algorithms that can bound the value of $\Delta$ in the fully-dynamic setting remains an interesting direction for further study.
>
> [1] Andrew Draganov, David Saulpic, and Chris Schwiegelshohn. Settling time vs. accuracy tradeoffs for clustering big data. In Proceedings of the ACM on Management of Data, 2024.
>
> [2] Vincent Cohen-Addad, et al. Tight FPT approximations for k-median and k-means. In Proceedings of the 46th International Colloquium on Automata, Languages, and Programming, 2019.
>
> [3] Di Wu, et al. Improved approximation algorithm for individual fairness k-median. In Proceedings of the International Conference on Combinatorial Optimization and Applications, 2024.

---

> > ### Author Rebuttal · Reviewer_YVKc · 2026-04-01
> >
> > Thanks for the rebuttal. After reading the rebuttals and reviews, I think the paper is strong and ICML is a good revenue with it. I keep my score.
> >
> > I do not think this publication requires an experimental section. There are many papers each year on $k$-clustering in ICML without experiments. I believe if the authors want to add an experimental section in the paper or appendix, it is better to withdraw the paper and resubmit it. That is, it is a major change and requires a level of review beyond the rebuttal.

---

### Decision · Program_Chairs · 2026-04-30

**Decision:**

Accept (regular)

**Comment:**

The paper studies the fully dynamic $k$-center problem with outliers and proposes a layered-sampling framework that removes the dependence on the aspect ratio $\Delta$. The reviewers agreed that this is a technically strong and well-written paper. While some concerns were raised regarding the increased approximation ratio, presentation clarity, and relevance to ML applications, the rebuttal addressed these points satisfactorily by clarifying the trade-offs, improving the exposition, and providing additional empirical evidence. In particular, reviewers acknowledged that eliminating the $\Delta$-dependence is a meaningful advance despite the trade-offs, and that the paper fits ICML. After the rebuttal, the scores are uniformly positive.